# Assimilating Compact Phase Space Retrievals (CPSRs): Comparison with Independent Observations (MOZAIC *in situ* and IASI Retrievals) and Extension to Assimilation of Truncated Retrieval Profiles

Arthur P. Mizzi[1], David P. Edwards[1], Jeffrey L. Anderson[2]

[1]National Center for Atmospheric Research, Atmospheric Chemistry Observations and Modelling Laboratory, Boulder, CO 80305, USA

[2]National Center for Atmospheric Research, Computational and Information Systems Laboratory, Boulder, CO, 80305, USA

*Correspondence to*: Arthur P. Mizzi (mizzi@ucar.edu)

## 1 Introduction

The adverse impacts of poor air quality on human health and welfare are well documented, e.g., Harvey (2016); Robichaud (2017). Air quality analyses and forecasts, more generally chemical weather products, are used to help understand and pre-empt poor air quality events. The accuracy of such chemical weather products depends in part on the application of chemical data assimilation to combine air quality observations with independent estimates of the air quality state to produce an "optimal" chemical weather analysis, Robichaud (2017). Air chemistry observations generally fall into two categories: *in situ* and remote. *In situ* observations come from direct observational platforms like samplers, and remote observation come from indirect observational platforms like satellites. Due to the spatial and temporal sparsity of *in situ* observations, air quality managers and researchers are increasingly relying on satellite observations. Such observations generally come in the form of "retrievals," and their use involves challenges that include: (i) low information density (the amount of information per retrieval is small), (ii) large volumes of data, (iii) incorporation of unobserved information from the retrieval prior, and (iv) correlated observation errors, Mizzi et al. (2016). In the chemical weather forecasting/data assimilation literature there have been several papers that have studied those challenges, see Joiner and Da Silva (1998), Migliorini et al., (2008), and Mizzi et al., (2016). Generally, other researchers have dealt with challenges (i) and (ii) by assimilating all the available retrievals, e.g., Jiang et al. (2015). They have dealt with challenge (iii) by assimilating the contribution from the retrieval priors, e.g., Jiang et al. (2015). And they have dealt with challenge (iv) by ignoring the error correlations, e.g., Barre et al. (2015). As discussed in Mizzi et al.

(2016), the problem with their approach for addressing challenges (i) and (ii) is that it is computationally expensive and
inefficient to assimilate all the retrievals. Some researchers have tried to address this by discarding (not assimilating) some of
the retrievals in the vertical profile, Arellano et al., (2007).  A similar strategy is used by some researchers to address biased
retrievals i.e., they do not assimilate the biased retrievals, Barre et al. (2015). Some of the results in this paper suggest there
are unexpected adverse impacts from discarding selected elements and assimilating the remaining elements of a retrieval
profile. Mizzi et al. (2016) introduced the assimilation of "compact phase space retrievals" (CPSRs) to address challenges (i)
and (ii) without discarding elements of the retrieval profile. In this paper, we extend the CPSR algorithm to truncated retrievals
profiles (retrieval profiles where some of the elements of the profile are not assimilated). However, as discussed herein, the
assimilation of truncated retrieval profile gives unexpected results due to role of the averaging kernel in the retrieval forward
operator.
Joiner and Da Silva (1998) was the first paper to address challenge (iii) – not assimilating the retrieval prior contribution.
They proposed three approaches.  In the first, they characterized the retrieval equation
$\boldsymbol{y}_r = \boldsymbol{A}\boldsymbol{y}_t + (\boldsymbol{I} - \boldsymbol{A})\boldsymbol{y}_a + \boldsymbol{\varepsilon}$ (1)
where $\boldsymbol{y}_r$ is the retrieval profile (column vector, dimension $n$ – the number of observations in a full retrieval profile), $\boldsymbol{I}$ is the
identity matrix (square matrix, dimension $n$ x $n$), $\boldsymbol{A}$ is the averaging kernel (square matrix, dimension $n$ x $n$, and rank $k$, where
$k < n$), $\boldsymbol{y}_a$ is the retrieval prior profile (column vector, dimension $n$), $\boldsymbol{\varepsilon}$ is the measurement error in retrieval space (column
vector, dimension $n$) with error covariance $\boldsymbol{E}_m$ (square matrix, dimension $n$ x $n$), and $\boldsymbol{y}_t$ is the unknown true atmospheric
profile (column vector, dimension $n$) as the sum of two linear transformations. The first transformation was a mapping of $\boldsymbol{y}_t$
to retrieval space by $\boldsymbol{A}$, and the second was a mapping of $\boldsymbol{y}_a$ to retrieval space by $\boldsymbol{I} - \boldsymbol{A}$. Then they projected $\boldsymbol{y}_r$ onto the
trailing left singular vectors from a Singular Value Decomposition (SVD) of $\boldsymbol{I} - \boldsymbol{A}$. In their second approach, they projected
$\boldsymbol{y}_r$ onto the trailing left singular vectors from an SVD of the retrieval prior error as mapped by $\boldsymbol{I} - \boldsymbol{A}$. Finally, their third
approach proposed a revised retrieval process that eliminated the need for $\boldsymbol{y}_a$. Those approaches were generally successful and
introduced the idea of assimilating phase space retrievals.  The second paper to address challenge (iii) was Migliorini et al.
(2008).  They formed the "quasi-optimal retrieval" (QOR) equation by subtracting the $(\boldsymbol{I} - \boldsymbol{A})$ term in Eq, 1 from $\boldsymbol{y}_r$ (to
remove the prior contribution). Then to address challenges (i), (ii), and (iv), they projected the result onto the leading left
singular vectors from an SVD of $\boldsymbol{E}_m$ and discarded those modes whose ensemble variance was much smaller than the
transformed observation error variance. Their approach was generally successful but did not address why the modes of the
observation error covariance should be related to the modes of the QOR. Finally, Mizzi et al. (2018) used QORs to address
challenge (iii) and two phase space transforms to address challenges (i), (ii), and (iv). The first was a compression transform
based on the leading left singular vectors of $\boldsymbol{A}$. This step enabled compression because $\boldsymbol{A}$ is highly rank deficient. Since those
singular vectors span the range of $\boldsymbol{A}$ and the QORs are in that range, their respective modes were mathematically related. The
second was a diagonalization transform to account for the observation error covariance during the assimilation. Their approach
was generally successful. The Mizzi et al. (2016) and Migliorini et al. (2008) algorithms are different. The Migliorini et al.
(2008) approach was motivated by rank deficiency of the observation error covariance and whether the phase space ensemble
error variance was small relative to the transformed observation error variance. The Mizzi et al. (2016) approach was motivated
by rank deficiency of the averaging kernel and accounting for the observation error covariance. The spaces spanned by the
respective transform vectors are different. The Migliorini et al. (2008) vectors spanned observation error covariance space,
and the Mizzi et al. (2016) vectors spanned QOR space. The Migliorini et al. (2008) compression was based on the relative
magnitude of the transformed ensemble error and observation error variance, and the Mizzi et al. (2016) compression was
based on the removal of redundant information for the QOR.
One aspect of assimilating retrievals not addressed by Migliorini et al. (2008) or Mizzi et al. (2016) is how to apply their
algorithms when the retrieval profile is truncated. Such an extension is necessary if one wants to assimilate only a portion of
the retrieval profile. Both methods can be extended so one goal of this paper is to document that extension for CPSRs and
evaluate the results.
Mizzi et al. (2016) demonstrated the utility of assimilating CPSRs by verifying the analysis and forecast results against the
assimilated observations. In this paper, we compare our results against both the assimilated and independent observations. As
in Mizzi et al. (2016), we assimilate conventional meteorological observations and Terra/Measurement of Pollution in the
Troposphere (MOPITT) CO retrievals, but here we also compare our analysis and forecast results with MetOp-A/Infrared
Atmospheric Sounding Interferometer (IASI) CO retrievals and Measurement of OZone, water vapor, carbon monoxide, and
nitrogen oxides by in-service AIrbus airCraft (MOZAIC) *in situ* CO profiles. Those comparisons are necessary because they
provide an independent assessment of the improved analysis fit and forecast skill. The remainder of this paper is organized as
follows: Section 2 describes the forecast/data assimilation system together with the assimilated meteorological and chemistry
observations. Section 3 describes the independent IASI and MOZAIC observations used in the verification analyses. Section 4
presents descriptions of our experiments, retrieval pre-processing methods, and extension of CPSRs to truncated retrieval
profiles. Section 5 compares the results of assimilating MOPITT CO retrievals (full and truncated profiles) with the IASI and
MOZAIC CO observations. Finally, Section 6 presents a summary of our results and conclusions.
**2 WRF-Chem/DART Regional Forecasting Ensemble Data Assimilation System: Set-Up and Assimilated Observations**
For the experiments reported here, we use the WRF-Chem/DART regional chemical transport/ensemble Kalman filter data
assimilation system introduced by Mizzi et al. (2016). WRF-Chem/DART is made up of the Weather Research and Forecasting
(WRF) model with chemistry (WRF-Chem) (www2.acd.ucar.edu/wrf-chem) coupled to the ensemble Kalman filter data
assimilation from the Data Assimilation Research Testbed (DART) (www.image.ucar.edu/DAReS/DART; Anderson et al.,
2009). WRF-Chem is a regional model that predicts conventional weather together with the transport, mixing, and chemical
transformation of atmospheric trace gases and aerosols. DART is an ensemble data assimilation system that uses the ensemble
adjustment Kalman filter of Anderson (2001, 2003) together with adaptive inflation and localization.
We conduct continuous cycling experiments with 6-hr cycling (00, 06, 12, and 18 UTC) for the period 1 June 2008 00 UTC
to 9 June 2008 18 UTC. To facilitate a large number of experiments, we use a reduced ensemble size of 20 members, a
horizontal resolution of 100km (101 x 41 grid points), and an abbreviated 9-day study period (compared to the 30-day period
used in Mizzi et al. (2016)). The reduced study period is not thought to negatively impact our results because the WRF-
Chem/DART spin-up occurs within the first 48 to 72 hours. The WRF-Chem domain extends from ~176 W to ~50 W and
~7 N to ~54 N. We use 34 vertical levels with a model top at 10 hPa and ~15 levels below 500 hPa. We use DART adaptive
prior covariance inflation with the recommended settings and DART Gaspari-Cohn localization with a localization radius half-
width of ~300 km in the horizontal. (Anderson, 2008). Vertical localization is not used. These are the same settings as used by
Mizzi et al. (2016).
The WRF-Chem initial and boundary conditions are derived from the National Oceanic and Atmospheric
Administration/National Center for Environmental Prediction (NOAA/NCEP) Global Forecast Model (GFS) 0.5º six-hour
forecasts. The WRF Preprocessing System (WPS) interpolates the GFS forecasts to our domain and generates the deterministic
boundary    conditions.    We    use    the    WRF    Data    Assimilation    System    (WRFDA)
(*http://www2.mmm.ucar.edu/wrf/users/docs/user_guide/users_guide_chap6;* Barker et al., 2012) to generate the initial
meteorological ensemble. The chemistry initial and boundary conditions are derived from the Model for Ozone and Related
Chemical Tracers: MOZART-4 (MOZART) forecasts, and WRF-Chem utilities are used to interpolate those forecasts to our
domain and generate the deterministic chemistry boundary conditions. The emissions and initial chemistry ensembles are
generated as described in Mizzi et al. (2016). The ensemble distributions are Gaussian with a specified mean and standard
deviation. The tails of those distributions are truncated to include 95% of the distribution and exclude outliers. That strategy
ensures that the emissions and initial chemistry variable concentrations are positive definite. We do not include horizontal
correlations for the emission perturbations because they are not relevant to the focus of this paper.
At each cycle time depending on the experiment, we assimilate conventional meteorological and chemistry observations with
DART and advance the analysis ensemble to the next cycle time with WRF-Chem. The resulting 6-hr forecast ensemble is
then used as the first guess in the next assimilation step. Our conventional meteorological observations are NCEP automated
data processing (ADP) upper air and surface observations (PREPBUFR observations), and our chemistry observations are
MOPITT CO mixing ratio retrieval profiles. MOPITT is an instrument on the National Aeronautics and Space Administration's
(NASA's) Earth Observing System Terra satellite. Its spatial resolution is 22 km at nadir over a swath width of 640 km. Its
thermal infra-red (TIR) measurements are sensitive to CO in the middle and upper troposphere, while its near infra-red (NIR)
measurements are sensitive to total column CO. MOPITT provides global coverage every three to four days. MOPITT CO is

reported on ten vertical levels starting at a variable surface pressure level and then ranging from 900 hPa to 100 hPa every 100 hPa. We assimilate the MOPITT V5 thermal-infrared/near-infrared (TIR/NIR) retrieval products described by Deeter et al. (2013). Validation results suggest that from 400 hPa to the surface the MOPITT CO retrievals are accurate to within 5%. Above 400 hPa, they may have a positive bias of ~14%, Deeter et al. (2013) and Martinez-Alonso et al. (2014), that has been addressed in subsequent MOPITT products, Deeter et al. (2014).

The horizontal resolution of the MOPITT data is much greater than that at which we run WRF-Chem. That difference translates to representativeness errors due to the smaller spatial scales that are resolved by the satellite but not by the model. To address those errors, we construct super-observations as follows: (i) sort the retrievals, retrieval priors, averaging kernels, and retrieval error covariances into bins that are ~90 km square, (ii) calculate the bin-average for each of those variables, and (iii) assimilate the bin-average retrievals. We use an arithmetic average (as opposed to an error covariance weighted average) when calculating the super-observations. We do not apply corrections to the retrieval error covariance super-observations because we are interested in the assimilation impact of the reported errors and can apply tuning to those errors and balance the root-mean square error (RMSE)/total spread fit as needed. Other studies e.g., Eskes et al. (2003), Miyazki et al. (2012 a and b, 2015), and Barre et al. (2016) have used similar super-observation strategies. We do not expect that tuning the observation errors would significantly impact our results because our diagnostic analyses showed that the RMSE and total spread were properly balanced.

**3 Independent Observations for Verification: MOZAIC *in situ* and IASI CO Retrieval Profiles**

In the first part of this paper, we compare the analysis and forecast results from assimilating MOPITT CO with independent observations (IASI CO retrievals and MOZAIC *in situ* CO profiles). IASI is an instrument on the EUMETSAT (European Organization for the Exploitation of Meteorological Satellites) polar orbiting MetOp-A satellite. Clerbaux et al. (2009). It measures temperature, water vapor, fractional cloud cover, cloud top temperature, ozone, carbon monoxide, and methane. IASI has been operating from 2006 to the present. Its mission is to provide observational support for numerical weather prediction. IASI measures CO radiances under cloud-free conditions with a horizontal resolution of 25 km over a swath width of ~2,200

km. IASI measurements are sensitive to CO in the mid- to lower troposphere. IASI provides global coverage every two days.
IASI CO is reported on 19 altitude levels ranging from the surface to 18 km every 1 km. Validation results suggest that the
CO retrievals are accurate to within 13%. For more information see www.eumetsat.int.
MOZAIC was a European Research Infrastructure (ERI) project that collected long-term, global-scale measurements of
atmospheric composition on international commercial airline flights from August 1994 to November 2014. Marenco et al.
(1998). MOZAIC collected *in-situ* measurement of ozone, water vapor, carbon monoxide, and total nitrogen oxides. The
available data products are geo-located (come with longitude, latitude, and pressure coordinates) and include simultaneous
meteorological observations. During MOZAIC, data acquisition was automatically performed on the ascent, descent, and
cruise phases of round-trip international flights between Europe and America, Africa, the Middle East, and Asia. For more
information see www.iagos.fr.
**4 Experimental Design**
We conduct WRF-Chem/DART forecast/assimilation cycling experiments that are similar to those of Mizzi et al. (2016). The
primary differences are the: (i) use of super-observations, (ii) extension of CPSRs to truncated retrieval profiles, and (iii) use
of localization to preclude the assimilated MOPITT CO observations from impacting any state variable other than CO. We
performed a control experiment where we assimilated only conventional meteorological observations (the MET experiment),
and we performed a series of chemical data assimilation experiments. In those experiments, we studied assimilation results
from four types of retrieval pre-processing strategies: (i) Volume Mixing Ratio retrievals (VMRRs, the associated experiment
is called the VMRR experiment), (ii) $Log_{10}$(VMRR) retrievals (L10VMRRs, the L10VMRR experiment), (iii) Compact Phase
Space Retrievals (CPSRs, the CPSR experiment), and (iv) Quasi-Optimal Retrievals (QORs, the QOR experiment). The CPSR
and QOR experiments (as applied to assimilation of full retrieval profiles) were studied by Mizzi et al. (2016). The VMRR
experiment and the L10VMRR and CPSR experiments as applied to assimilation of truncated retrieval profiles are new. We
include the L10VMR and QOR experiments as applied to retrieval full profiles because, as discussed in the Introduction, our
comparison of those experiments with independent observations (discussed below in Section 5.1) suggests that it may be

beneficial to not assimilate MOPITT CO retrievals in the upper troposphere due to their possible bias. That concern motivates application of the L10VRR and CPSR experiments to the assimilation of truncated retrieval profiles. The rest of this section describes those experiments. It should be noted that the different retrieval pre-processing methods (making up the different experiments) are applied after the customary quality assurance/quality control (QA/QC) checks that might discard entire retrieval profiles. Those forecast/assimilation experiments are summarized in Table 1.

## 4.1 The VMRR and L10VMRR Experiments

The MOPITT CO retrieval, averaging kernel, and error covariance products are reported in units of $\log_{10}(VMR)$. The IASI CO products are in VMR. For ease of comparison and interpretation, it is convenient to convert the MOPITT data from L10VMRR to VMRR. While it is possible to convert the retrievals and error covariance, it is not possible to convert the averaging kernels. Consequently, for the VMRR experiment the DART forward operator for MOPITT CO converts the state space CO profile from VMRRs to L10VMRRs, applies the averaging kernel, and then converts the resulting expected observation (the expected retrieval profile) to VMR. For the L10VMRR experiment a conversion is not necessary because the state space CO profile is in $\log_{10}(VMR)$. Conceptually, we expect little difference between the VMRR and L10VMRR experiments due to an underlying assumption that L10VMRRs have a Gaussian distribution and the VMRRs have a lognormal distribution (Deeter et al., 2007). However, non-linearity of the base-ten exponential operator that relates the L10VMRRs to the VMRRs and the extent to which the VMRR distributions are non-Gaussian may introduce differences. So, one goal of the related experiments is to determine whether those differences are significant. Another reason is to include pre-processing methods that enable us to not assimilate selected retrievals so we can compare the assimilation/forecast results with those from applying CPSRs to truncated retrieval profiles.

## 4.2 The QOR Experiment

The assimilation of QORs was discussed in Mizzi et al. (2016). We include QOR assimilation/forecast experiments for completeness and to provide a reference against which to compare the other retrieval pre-processing experiments. In addition (although not discussed herein), QOR pre-processing can be applied to truncated retrieval profiles using the extension discussed in the next section on the CPSR experiment.

QORs are retrieval residuals introduced by Migliorini et al. (2008). They are derived by writing the retrieval equation as
$$\boldsymbol{y}_r - (\boldsymbol{I} - \boldsymbol{A})\boldsymbol{y}_a - \boldsymbol{\varepsilon} = \boldsymbol{A}y_t. \tag{2}$$
and transforming Eq. (2) with the left singular vectors from the SVD of $\boldsymbol{E}_m$ divided by the square root of the associated singular
value. If the SVD of $\boldsymbol{E}_m$ is $\boldsymbol{E}_m = \boldsymbol{\phi}\boldsymbol{\sigma}\varphi^T$, then the QOR profile is defined as
$$\sigma^{-1/2}\boldsymbol{\phi}^T(\boldsymbol{y}_r - (\boldsymbol{I} - \boldsymbol{A})\boldsymbol{y}_a - \boldsymbol{\varepsilon}) = \sigma^{-1/2}\boldsymbol{\phi}^T\boldsymbol{A}y_t \tag{3}$$
and the transformed $\boldsymbol{E}_m$ is the identity matrix. That transform is similar to the CPSR diagonalization transform described in
the next section except Migliorini et al. (2008) applied the QOR transform to the raw averaging kernel and the raw error
covariance while Mizzi et al. (2016) applied it to the compressed averaging kernel and the compressed error covariance. In
our application of QORs, there is no filtering of the dominate modes. Also, in general the QOR transform has no zero singular
values because $\boldsymbol{E}_m$ is not singular.
**4.3 The CPSR Experiment and the Extension of CPSRs to Assimilation of Truncated Retrieval Profiles**
The derivation and assimilation of CPSRs was introduced by Mizzi et al. (2016). They derived CPSRs by applying two
transforms to Eq. 2: (i) a compression transform based on the SVD of $\boldsymbol{A}$, and (ii) a diagonalization transform based on the SVD
of the compressed $\boldsymbol{E}_m$. Their application can be characterized as CPSRs applied to full retrieval profiles (because none of the
elements in the retrieval profile were discarded) or to square systems (because $\boldsymbol{A}$ is a square matrix). If we discard one or more
elements of $\boldsymbol{y}_r$, then we must also discard the corresponding rows of $\boldsymbol{A}$ (call the modified forms $\widehat{\boldsymbol{y}}_r$ and $\widehat{\boldsymbol{A}}$ respectively)**.** The
resulting $\widehat{\boldsymbol{A}}$ is not a square matrix. Note that we must also discard the corresponding rows and columns of $\boldsymbol{E}_m$, so it remains
square but its dimension is reduced. This application can be characterized as CPSRs applied to truncated retrieval profiles
(because some of the elements of the retrieval profile have been discarded) or to rectangular systems (because $\widehat{\boldsymbol{A}}$ is a non-
square rectangular matrix). The mathematical formalism for CPSRs applied to rectangular systems is the same as that for
square systems because Mizzi et al. (2016) used a SVD (as opposed to an eigenvalue decomposition) in their derivation. In the
remainder of this section, we extend the derivation of CPSRs from Mizzi et al. (2016) to rectangular systems.
We begin by conceptually discarding $q$ elements of $\boldsymbol{y}_r$. Generally, we discard the elements of the full retrieval profile $\boldsymbol{y}_r$ that
are known to be systematically bad observations. If we discard multiple elements, they need not be sequential. The resulting
truncated retrieval profile is denoted $\hat{\boldsymbol{y}}_r$ and its dimension is $\hat{n} = n - q$. We must also discard: (i) the corresponding elements
of $\boldsymbol{\varepsilon}$ to get $\hat{\boldsymbol{\varepsilon}}$ with dimension $\hat{n}$, (ii) the corresponding rows of $\boldsymbol{A}$ to get $\widehat{\boldsymbol{A}}$ with dimension $\hat{n} \times n$, and (iii) the corresponding
rows and columns of $\boldsymbol{E}_m$ to get $\widehat{\boldsymbol{E}}_m$ with dimension $\hat{n} \times \hat{n}$. Without loss of generality, we can drop the ˆ notation for the
remainder of this paper and let $\boldsymbol{y}_r$, $\boldsymbol{\varepsilon}$, $\boldsymbol{A}$, and $\boldsymbol{E}_m$ represent their respective terms before and after discarding the retrieval
elements that will not be assimilated. The rest of the derivation is the same as in Mizzi et al. (2016).
First, we apply the compression transform based on the leading left singular vectors of $\boldsymbol{A}$. If $\boldsymbol{A} = \boldsymbol{USV}^T$ is the SVD and
$A_0 = U_0 S_0 V_0^T$ is the truncated SVD where the trailing singular vectors (those whose singular values are less than an *ad hoc*
threshold of $1.0 \times 10^{-4}$) are replaced with zero vectors and the trailing singular values are set to zero, then the compressed
form of Eq. 2 is
$$\boldsymbol{U}_0^T(\boldsymbol{y}_r - (\boldsymbol{I} - \boldsymbol{A})\boldsymbol{y}_a - \boldsymbol{\varepsilon}) = \boldsymbol{S}_0 \boldsymbol{V}_0^T \boldsymbol{y}_t \tag{4}$$
and the compressed error covariance is
$$\boldsymbol{U}_0^T \boldsymbol{E}_m \boldsymbol{U}_0. \tag{5}$$
In that step, there is no filtering of the dominate modes. Next, we apply the diagonalization transform. If the SVD of the
compressed error covariance in (5) is $\boldsymbol{U}_0^T \boldsymbol{E}_m \boldsymbol{U}_0 = \boldsymbol{\Phi}\boldsymbol{\Sigma}\boldsymbol{\Psi}^T$, then the diagonalized and conditioned form of Eq. 4 is
$$\boldsymbol{\Sigma}^{-1/2}\boldsymbol{\Phi}^T\boldsymbol{U}_0^T(\boldsymbol{y}_r - (\boldsymbol{I} - \boldsymbol{A})\boldsymbol{y}_a - \boldsymbol{\varepsilon}) = \boldsymbol{\Sigma}^{-1/2}\boldsymbol{\Phi}^T\boldsymbol{S}_0\boldsymbol{V}_0^T\boldsymbol{y}_t \tag{6}$$
and that of (5) is the identity matrix. Eqs. 4 – 6 and the fully transformed error covariance are the same as in Mizzi et al. (2016)
except that unwanted retrieval elements have been discarded.
Finally, we note that the rank of $\boldsymbol{A}$ and the rank of $\widehat{\boldsymbol{A}}$ are generally the same provided the difference between the dimension of
$\boldsymbol{A}$ and the rank of $\boldsymbol{A}$ is greater than or equal to the number of discarded elements from the retrieval profile i.e., $n - k \geq q$.
That statement is not necessarily true, but given the rank deficiency of $\boldsymbol{A}$ it is usually true. We also note that the $\boldsymbol{\Sigma}^{-1/2}\boldsymbol{\Phi}^T\boldsymbol{S}_0\boldsymbol{V}_0^T$
on the right side of Eq. 6 is the transformed averaging kernel. It represents the sensitivity of the phase space retrievals (the
CPSRs) to the true CO concentrations at each vertical level. Unlike the raw averaging kernel, which included sensitivities to
the null space contributions to the retrieval (the linearly dependent contributions from the right side of Eq. 2), the transformed
averaging kernel contains only sensitivities for the measurement contributions to the retrieval (the linearly independent
contributions from the right side of Eq. 2).
**5 Results**
**5.1 Assimilation of Full Retrieval Profiles**
In this section, we look at assimilation/forecast results from the experiments described in Section 4. The reader should note
that the CPSR and QOR experiments are the same as the MOP CPSR and MOP QOR experiments from Mizzi et al. (2016)
except: (i) the study period is shorter (nine days as opposed to one month), (ii) we assimilate MOPITT super-observations, and
(iii) we use localization to preclude the assimilated MOPITT CO observations from impacting any state variable other than
CO.
Figure 1 show forecast verification statistics (RMSE and Bias) for the different experiments when compared against the
assimilated MOPITT CO retrievals on the left and the independent IASI CO retrievals on the right. For the MOPITT
comparison, the MOPITT CO forward operator has been applied to the WRF-Chem results so the comparison is made in
MOPITT CO retrieval space.  Similarly, for the IASI comparison the IASI CO forward operator has been applied so the
comparison is made in IASI CO retrieval space.  The left panel can be compared with Fig. 8 from Mizzi et al. (2016).
Qualitatively, that comparison shows that the two figures are similar. The MET experiment yields the highest RMSE and bias
while the CPSR and QOR experiments yield lower RMSE and bias. Similar results are seen in the IASI CO comparison. It is
interesting that for both comparisons: (i) The VMRR experiment shows a slight degradation when compared to the MET
experiment, and (ii) the VMRR and L10VMRR experiments are similar to the MET experiment. We suspect that result (i) is
a consequence of the non-linearity of the base-ten log function and the non-Gaussianity of the VMRR distributions, and
result (ii) is a consequence of the magnitude of the observation errors used in the VMRR and L10VMRR experiments
(discarding the observation error cross-covariance produced observation error variances that are large compared to those
produced by the CPSR diagonalization transform) and the length of the study period. We believe the CPSR observation errors
are smaller due to the compression step of the CPSR transform. They cannot be smaller due to the diagonalization step because
that is a variance maximizing rotation. So, if the compression had no filtering effect on the errors, the variance resulting from
the diagonalization step would no smaller than that from the compression step. One consequence of relatively large observation
errors is that it takes more cycles for the assimilation to show an impact. We have run similar experiments with a longer study
period and found assimilation impacts. We do not view that as a deficiency in the experimental design. We are interested in
the assimilation of CPSRs. If they show an impact during a shorter study period but more conventional methods that do not
account for redundant information or error correlations fail to show an impact, then that failure identifies deficiencies in the
conventional methods.
Figure 1 generally shows increasing improvement when moving from the MET to L10VMRR to CPSR and QOR experiments.
As discussed previously the VMRR and L10VMRR experiments show little to no improvement over the MET experiment. In
Fig. 1 the CPSR and QOR experiments show comparable skill. That result can also be seen in Mizzi et al. (2016) by comparing
Figs. 3 and 7. There are two potential explanations. First, we use the retrieval space retrieval error covariance ($E_r$) as the
observation error covariance to account for other unquantified error sources, and $E_r = (I - A)E_a$ where $E_a$ is the retrieval *a*
*priori* error covariance. If the singular vectors of $E_r$ are equivalent to those of $A$, we would get similar results from the CPSR
and QOR experiments. However, $E_a$ is specified in the retrieval algorithm as a covariance matrix, and generally there is no
reason to suspect that the singular vectors of $E_r$ are equivalent to those of $A$ (for MOPITT CO they are not equivalent because
their respective singular vectors are not orthogonal). Second, in the QOR experiment the diagonalization transform rotates the
QOR equation so that the observation error cross-covariance contributions for each mode are included in their corresponding
observation error variance. However, those modes are linearly dependent in the space defined by the rotated averaging kernel
because the rotated averaging kernel is still singular. When those linearly dependent modes are assimilated, there is very little
adjustment to the analysis. Consequently, the CPSR and QOR experiments yield similar results because: (i) the QOR
experiment apportions the error and assimilates the linearly dependent modes (which have little or no impact), while (ii) the
CPSR experiment apportions the error and does not assimilate the linearly dependent modes. Those results differ from the
VMRR and L10VMRR experiments because the observation error variance used in the retrieval space experiments does not
account for the error cross-covariance contributions, and the linearly independent portion of that error is different from that in
the CPSR and QOR experiments.
In Fig. 2, we compare results from the CPSR and MET experiments with the MOZAIC ascent and descent soundings for
Dallas, TX (two soundings composited), Portland, OR (four soundings composited), and Philadelphia, PA (two soundings
composited). The MOZAIC soundings from 1 June 2008 (Dallas, TX) were discarded because they were observed during our
spin-up period.  Otherwise, out MOZAIC comparisons were not impacted by forecast/assimilation system spin-up. No other
MOZAIC soundings were available for our study period and domain.  The MOZAIC soundings used in Fig. 2 were generally
not spatially (within several hundred kilometres) or temporally (within three hours) coincident with the MOPITT observations.
We linearly interpolated the WRF-Chem forecasts to the MOZAIC observation times and locations and then composited the
results. We did not plot the composited MOZAIC profile below 750 hPa because those data are more representative of the
lower troposphere over urban areas than are our model grid and assimilated super-observations.  The MOZAIC comparison
results are qualitatively similar to those from Fig. 1. The CPSR experiment shows that: (i) assimilation of phase space
retrievals improves the 6-hr forecast skill in the middle and lower troposphere when compared to the MET experiment for
Dallas, TX  and Portland, OR but degrades the skill in the upper troposphere, (ii) assimilation generally degrades skill
throughout the troposphere for Philadelphia, PA, (iii) none of the assimilation impacts are significant based on the ensemble
variability, and (iv) assimilation provides little or no change near the surface.  The upper tropospheric degradation in results (i)
and (ii) is related to the positive bias in upper tropospheric MOPITT retrievals discussed earlier. Result (iii) is likely a result
of the small sample size, but given the magnitude of the skill differences in the middle and upper troposphere and the "near-
significance" suggested by some of the error bars, we think there is value in presenting these results. Result (iv) is somewhat
unexpected because MOPITT retrievals are documented to have sensitivity to CO in the upper and lower troposphere (Deeter
et. al. 2007). Also, other chemical data assimilation researchers, e.g. Jiang et al. (2013) and Barre et al. (2015), have
reported near-surface improvements due to assimilation of MOPITT CO multi-spectral retrievals. We suspect result (iv)
occurs because MOPITT's upper tropospheric sensitivities dominate its lower tropospheric sensitivities in the transformed
system.
To test that hypothesis, we plot a histogram of the MOPITT degrees of freedom for signal (DOFS) for all terrestrial profiles
in our domain during the study period in Fig. 3. The MOPITT DOFS is a measure of the amount of independent observed
information in a retrieval profile. If a profile contains independent information from the upper and lower troposphere, its DOFS
must be ~2.0. The central histogram of Fig. 3 shows that the mean, median, and mode of DOFSs during this period are ~1.5
and that DOFSs greater than ~2.0 are relatively rare (< 5%). To gain a better understanding of the vertical structure of the
MOPITT retrieval information content, we present a composite analysis for averaging kernel profiles in the neighborhood of
different DOFS values in the lower row of Fig. 3 where panel (a) is the composite averaging kernels for all DOFS, (b) is for
(0.9 < DOFS < 1.1, ~10% of the histogram probability mass), (c) is for (1.4 < DOFS < 1.6, ~26%), and (d) is for (1.9 <
DOFS < 2.1, ~4%). Those panels show that the dominant sensitivity appears to be to the upper troposphere and that as
the DOFS approaches 2.0 the sensitivity to the lower troposphere increases. That sensitivity distribution could explain
the improvement drop off in the lower troposphere for the MOZAIC comparisons because retrievals with sensitivity to
the lower troposphere are relatively rare. However, linear dependencies in the composite averaging kernels of Fig. 3
can mask the significance of the sensitivities to the lower troposphere in the more common DOFS categories.
To unmask those sensitivities, Fig. 4 presents a composite analysis of the different DOFS sensitivities based on the CPSR
compression and diagonalization transforms, and Table 2 presents the total and modal information content associated
with Fig. 4. The upper row of Fig. 4 shows composite vertical profiles of the leading left singular vectors of the averaging
kernel. Those singular vectors: (i) span the range of the averaging kernel (the QOR space), (ii) are ranked such that the
first singular vector explains the greatest amount of vertical variability in the QOR profile, the second singular vector
explains the next greatest amount of variability, and so forth, and (iii) have arbitrary sign, so we chose the sign that has
the greatest physical meaning, i.e., we apply a -1.0 scaling to the first and second rows of Fig. 4. Table 2 shows that for
$0.9 \leq DOFS \leq 1.0$ most of the information is in the first mode, for $1.4 \leq DOFS \leq 1.5$ two-thirds of the information is in the
first mode and one-third is in the second mode, and for $1.9 \leq DOFS \leq 2.1$ one-half of the information is in the first mode
and one-half is in the second mode. In Fig. 4, we retained three singular vectors for completeness, but it should be
remembered the third vector (and sometimes the second vector) may map information to the null space of the
transformed averaging kernel. The second row of Fig. 4 shows composite vertical profiles for the compressed averaging
kernels. These profiles show the vertical sensitivity of compressed QORs to the true atmospheric state. The bottom row
shows the composite vertical profiles for the compressed and rotated averaging kernels (the profiles after the full CPSR
transformation). These profiles show the vertical sensitivity of CPSRs to the true atmospheric state.
Figure 4 shows some interesting results. The upper row of Fig. 4 shows that for DOFS ≈ 1.0 (column (b)) the first leading
singular vector has positive sensitivity near the surface and negative sensitivity in the middle to upper troposphere
(remember that the second and third leading vectors may map to the null space for DOFS ≈ 1.0). As the DOFS increases
to 1.5, the first and second leading vectors have positive sensitivity near the surface and weakly negative sensitivity in
the middle to upper troposphere, and for DOFS of 2.0, the first leading vectors has positive sensitivity throughout the
troposphere while the second leading vectors has positive sensitivity near the surface and negative sensitivity in the
middle to upper troposphere.
After applying the CPSR diagonalization transform, the DOFS-dependent sensitivity patterns in the second row of Fig. 4
change, and the final patterns (those of the compressed averaging kernels) are shown in the bottom row. These profiles
show that for all DOFS (column (a)) the first leading mode has its greatest sensitivity near the surface and the sensitivity
decreases to a near-zero positive minimum in the upper troposphere.  Similarly, the second leading mode has it greatest
positive sensitivity near the surface but has strong negative sensitivity in the upper troposphere.  The right three
columns of the second row in Fig. 4 show the dependency of the vertical sensitivity on the DOFS for the compressed
QORs. As seen with the singular vectors, as the DOFS increases the sensitivity changes from weak positive sensitivity
near the surface and strong negative sensitivity in the upper troposphere to strong positive sensitivity throughout the
troposphere for the first leading mode and positive sensitivity near the surface and strong negative sensitivity in the
upper troposphere for the second leading mode. Those results suggest that the MOPITT retrievals (and therefore the
results in Fig. 2) should be sensitive to CO in the lower troposphere/near the surface. However, an interesting thing
happens when we account for the reported retrieval error covariance. The lower row of Fig. 4 shows the compressed
and rotated averaging kernel profiles, which account for that error covariance. Here the negative scaling cancels each
other because the SVD has been applied twice. These results show first that the significance of the leading modes
becomes reversed due to diagonalization transform and scaling by the inverse square root of the compressed and
rotated error variance. This does not mean that the third leading mode from the first two rows of Fig. 4 becomes a
dominant mode because it may still be mapping to the null space, i.e., the leading CPSR modes (those with the smaller
observational error variance) may be mapping to the null space, and the trailing CPSR modes are mapping to the
domain of the transformed averaging kernel. That suggests that there may be benefit to not assimilating some of the
leading CPSR modes which would be similar to not assimilating the phase space modes with small observational error
as was done by Migliorini et al. (2008). The bottom row of Fig. 4 shows that after removing the linear dependencies
and accounting for the observation errors, the compressed and rotated averaging kernel has its greatest sensitivity in
the upper troposphere for DOFS < 2.0 and weakest sensitivity near the surface for DOFS $\approx$ 2.0.  That explains why our
comparison of the CPSR experiment with the MOZAIC observations in Fig. 2 did not show assimilation impacts near the
surface.  Other researchers who have assimilated MOPITT CO could not have found this result because they did not
adjust for the averaging kernel linear dependencies or for the observation error covariance.  See e.g., Jiang et al. (2013)
and Barre et al. (2015).
Figures 5 and 6 show contour maps comparing the MET and CPSR experiments for 9 June 2008 18 UTC (Fig. 5) as well as
the assimilated MOPITT and independent IASI CO retrievals (Fig. 6). Examination of the forecast maps in the upper panel
and the forecast difference map (CPSR experiment minus MET experiment) in the lower left panel of Fig. 5 (defined as CPSR
EX CO Del-Fcst) shows that assimilation of MOPITT CO retrievals increased the CO concentrations over some areas (southern
California, southern Baja, and northern Atlantic east of New England) and decreased the concentrations over broader areas
(mid- to northeastern United States, southeastern United States, and southern Gulf of Mexico). Comparison of the MOPITT
CO retrievals in the upper panels of Fig. 6 (the assimilated retrievals) with Fig. 5 shows that the analysis and forecast impacts
are generally consistent with the observations. Over southern Baja the MOPITT observations in Fig. 6 report CO on the order
of 50 ppb while the forecast in Fig, 5 (the assimilation prior) reports CO on the order of 100 ppb.  The assimilation increment
shows a CO reduction (consistent with the MOPITT observations) on the order of 50 ppb.  Similarly, the increased CO in the
central United States, over Kansas and Nebraska and in the southeastern United States near Georgia, South Carolina, and
Virginia (highlighted by the analysis increment map in the lower right panel of Fig. 5) is consistent with relatively low CO in
the prior when compared to the MOPITT observations. Comparison of the analysis increments, the assimilated MOPITT CO
retrievals, and the independent IASI CO retrievals (lower panels of Fig. 5 and Fig. 6) confirms that the assimilation of MOPITT
retrievals generally improved the analysis and forecast agreement with the IASI retrievals compared to the MET experiment.
Over Baja MOPITT and to a lesser extend IASI in Fig. 6 report CO on the order of 50 ppb to 75 ppb.  The assimilation prior
(the CO forecast) in Fig. 5 has CO on the order of 125 ppb to 150 ppb.  The corresponding increment is a CO reduction on the
order of 50 ppb.  The IASI CO map in Fig. 5 also confirm adjustments over Oklahoma, Kansas, and Nebraska, and to a lesser
extent to the east of Georgia, South Carolina, and Virginia.
Figure 7 shows horizontal domain average vertical profiles for the MET and CPSR experiments compared against horizontal
domain average profiles for MOPITT and IASI. The WRF-Chem profiles are plotted in retrieval space (after accounting for
the averaging kernel and assimilation prior). Comparison of the model and MOPITT profiles (left two panels of Fig. 7) shows
that the CPSR experiment generally draws the forecast and analysis profiles closer to MOPITT than does the MET experiment.
The error bars are based on the ensemble uncertainty and suggest that those improvements are significant throughout the
troposphere. The same comparisons with the IASI profiles (right two panels of Fig. 7) shows a different result: (i) in the upper
(pressure ($p$) < 250 hPa) the MET experiment draws the forecast and analysis profiles closer to IASI than does the CPSR
experiment, and (ii) for $p$ > 250 hPa) the CPSR experiment draws the profiles closer to IASI. Here again, the error bars suggest
that those changes are significant throughout the troposphere. The results from the comparison with IASI highlight the
problem, previously discussed for the MOZAIC comparisons in Fig. 2, with assimilating the potentially biased MOPITT CO
retrievals. To address that problem, we propose to discard the biased retrievals and assimilate the unbiased truncated retrieval
profiles with the extended CPSR method described in Section 4.
In summary, this section shows that assimilation of MOPITT CO retrievals improves analysis fit and forecast skill when
compared to MOPITT as well as when compared to the independent (not assimilated) IASI and MOZAIC observations. It
shows that: (i) the CPSR experiment improves the skill when compared to assimilation of raw retrievals (VMRR and
L10VMRR) because the phase space transformation reduces the phase space observation errors, and (ii) the CPSR and QOR
experiments yield similar results because they account for the observation error cross-covariance contribution in the same way
(the diagonalization transform) and because the linearly dependent portion of the transformed retrievals do not contribute to
the analysis increment (explicitly with CPSRs and implicitly through the assimilation algorithm for compressed QORs). It also
shows that the CPSR experiment did not improve the skill in the lower troposphere near the surface because: (i) MOPITT CO
profiles with sufficient DOFS to resolve the lower tropospheric CO signal are relatively rare (for this domain and study period),
and (ii) an analysis of the impact of the CPSR compression and diagonalization transforms shows that the upper tropospheric
CO signal dominates the MOPITT CO sensitivities. Finally, this section shows that in the upper troposphere assimilation of
biased MOPITT observations introduced analysis and forecast error relative to the IASI observations.
**5.2 Assimilation of Truncated Retrieval Profiles**
In this section, we test two methods for assimilating truncated retrieval profiles: (i) assimilate L10VMRR retrievals after
discarding the biased retrievals (the L10VMRR-RJ3 experiment where the RJ3 indicates that we do not assimilate retrievals
above 300 hPa – the upper three levels of the MOPITT CO retrieval profile) and (ii) assimilate CPSRs with the extension to
truncated retrieval profiles as described in Section 4.3 (the CPSR-RJ3 experiment). The L10VMRR-RJ3 experiment is
included only for comparison purposes. If the L10VMRR-RJ3 and CPSR-RJ3 experiments give similar results then the CPSR-
RJ3 approach is preferred because it is computationally less expensive, removes linear dependencies, and accounts for the
observation error covariance.
Figure 8 shows vertical profiles for the L10VMRR-RJ3 and CPSR-RJ3 experiments with results from the full retrieval profile
assimilation experiments included for reference. In these experiments, we are assuming that: the MOPITT retrievals are
positively biased in the upper troposphere and the IASI CO retrievals more accurately reflect the true atmospheric state.
Comparisons against the assimilated MOPITT observations in the upper panels show that discarding the biased observations
had the desired effect – in the upper troposphere the L10VMRR-RJ3 experiment removes the bias and the analysis profile is
drawn closer to that of the MET experiment than in the L10VMRR experiment. Similar results are seen for the CPSR-RJ3
experiment in the last two columns of the upper row. Unexpectedly, for both experiments, not assimilating observations in the
upper troposphere had a negative impact in the lower troposphere. A comparison with IASI CO retrievals in the lower row of
Fig. 8 shows similar results: (i) the L10VMRR-RJ3 and CPSR-RJ3 retrieval space profiles are drawn closer to the IASI profile
than the L10VMRR and CPSR profiles in the upper troposphere, and (ii) the skill is degraded in the middle and lower
troposphere. We investigate the cause of those lower tropospheric results later in this section, but first we review the horizontal
impacts of the truncated retrieval assimilation experiments.
Figures 9 and 10 show contour maps for the CPSR-RJ3 experiment. Figure 9 shows the near-surface impacts, and Fig. 10
shows the upper tropospheric impacts. The CO 6-hr forecast contour maps in the upper row of Fig. 9 confirm that not
assimilating the biased retrievals negatively impacted the lower troposphere because the assimilation impacts are small. The
forecast difference maps in the lower row show the impacts in the lower troposphere from assimilating MOPITT CO in the
upper troposphere.  The CPSR-RJ3 experiment does not have those impacts. It has small large-scale CO decreases over the
oceans and eastern United States similar to but weaker than in the CPSR experiment. Also, the magnitude of positive forecast
differences at CO hot spots over Southern California, Baja, and the northeastern United State has decreased. Figure 10 shows
fewer large scale changes for the CPSR-RJ3 experiment except for the reductions over the southeastern United States and Gulf
of Mexico. Here the CPSR-RJ3 experiment has large reductions in the CO adjustments (reducing the bias). Figure 10 provides
additional demonstration that discarding the biased retrievals reduces the model's upper tropospheric bias.  Unfortunately, we
obtain that result at the expense of reduced improvements in the lower troposphere.
A verification analysis for the L10VMRR-RJ3 and CPSR-RJ3 experiments is presented in Fig. 1. The L10VMRR-RJ3 and
CPSR-RJ3 experiments have degraded forecast skill compared to the full profile assimilation experiments (the CPSR and QOR
experiments), but the CPSR-RJ3 experiment has slightly improved skill compared to the L10VMRR-RJ3 experiment. That
small improvement is likely due to observation error covariance reductions from the CPSR transform as discussed earlier.
In summary, not assimilating the biased observations had positive impacts in the upper troposphere and negative impacts in
the middle to lower troposphere. We suspect the negative results occurred for two reasons. Discarding retrievals and their
averaging kernels: (i) reduces the total information content of the assimilated retrievals so that the assimilation adjustments
are small; and (ii) reduces the sensitivity of the transformed averaging kernel so that the expected retrievals are less sensitive
to the true atmospheric profile. Those reductions combine to reduce the ensemble state variable correlations and consequently
the assimilation impacts. To test explanation (i) we compare the trace of the composited raw averaging kernel for the CPSR
experiment with that for the CPSR-RJ3 experiment. The results are shown in the first two rows of Table 3 where "Full Profile"
is from the CPSR experiment, and "Reject Top Three" is from the CPSR-RJ3 experiment. Comparison of those results shows
a 25% reduction in the trace indicating that the total information content of the assimilated retrievals for the CPSR-RJ3
experiment is 25% less than that for the CPSR experiment. For comparison purposes, Table 3 also shows trace reductions from
not assimilating retrievals in the middle troposphere (23% reduction) and lower troposphere (9% reduction). Those results
suggest that most of the information in the MOPITT CO retrievals is from the upper troposphere, the second greatest amount
is from the middle troposphere, and the smallest amount is from the lower troposphere. To test explanation (ii) we plot the
compressed and fully transformed averaging kernels in Fig. 11 where column (a) is for the CPSR experiment and column (b)
is for the CPSR-RJ3 experiment. Figure 11 is similar to the last two rows of Fig. 4. Recall that the first row represents the
sensitivity of the compressed QORs to the true CO concentrations, and the second row represents the sensitivity of the CPSRs
to the true CO concentrations. Comparison of columns (a) and (b) shows that for the CPSR-RJ3 experiment, the leading mode
sensitivities are reduced when compared to the CPSR experiment. The state variable correlations are proportional to those
sensitivities, so the reduced correlations result in analysis increment reductions. For comparison purposes columns (c) and (d)
of Fig. 11 show results from experiments that discard retrievals in the middle and lower troposphere. Those profiles, in
combination with Table 2, show that most of the information and sensitivity is associated with the upper and mid-tropospheric
retrievals. Discarding upper tropospheric retrievals alters the sensitivity magnitudes while discarding middle tropospheric
retrievals alters the magnitudes and vertical structure. One interesting result is that most of the sensitivity loss in column (c) -

the "Reject Middle Three" experiment - appears to be associated with the CPSR diagonalization transform. That suggests that the sensitivity loss is dependent on specification of the retrieval *a priori* error covariance.

Those changes occur because as different rows of the averaging kernel are discarded: (i) the amount of observed information in the modified averaging kernel changes, and (ii) the vertical structure of the bases for the range and domain of the modified averaging kernel changes. The impact of changes in the information content in point (i) were discussed earlier. The impact of changes to the bases in point (ii) has important consequences. The leading left singular vectors of the transformed averaging kernel span the range of the transformed averaging kernel but their vertical structure and possibly their dimension change when retrievals are discarded. That means the phase space observations change because the basis vectors used in the compression transform are different, and their sensitivity to the truncated retrieval profile vector is different. Similarly, the leading right singular vectors of the transformed averaging kernel span the domain of the transformed averaging kernel, but their vertical structure changes when retrievals are discarded. Those changes occur solely because the information content of the transformed averaging kernel is reduced (since the dimension of its domain – the space where the true CO profiles reside – is unchanged). Those changes are significant because they alter the elements (or levels) of the true profile to which the transformed averaging kernel is sensitive. To summarize not assimilating elements of the full retrieval profile alters the levels of the retrieval profile to which the phase space observations are sensitive. Discarding those elements also alters the levels of the true CO profile to which the transformed averaging kernel is sensitive. Those sensitivity changes occur regardless of whether the assimilation is done in phase space as in the CPSR–RJ3 experiment or in retrieval space as in L10VMRR–RJ3 experiment. Consequently, results from the L10VMRR-RJ3 and CPSR-RJ3 experiments are similar.

This section shows that CPSRs can be extended to the assimilation of truncated retrieval profiles but that discarding upper tropospheric observations for MOPITT significantly reduces the total information content of the assimilated observations and the vertical sensitivity of the transformed averaging kernel profiles. Those reductions translate to reductions in the state variable correlations and commensurate reductions in the analysis increments. We are studying modification of the CSPR extension to truncated retrieval profiles to address the non-local impacts.

**6 Summary and Conclusions**
This paper had two goals: (i) compare the results from assimilating CPSRs with independent observations (we used MOZAIC
*in situ* observations and IASI CO retrievals as the independent observations), and (ii) extend CPSRs to the assimilation of
truncated retrieval profiles. The comparison with independent observations showed that: (i) assimilation of raw retrievals
(VMRRs and L10VMRRs) had little impact on the analysis fit and forecast skill due to the magnitude of the observation errors
and the length of the study period, and (ii) the assimilation of phase space retrievals (CPSRs and QORs) improved both fit and
skill. Conceptually, we expect the assimilation of raw retrievals and phase space retrievals to yield similar results. However,
phase space transformation of the observation error covariance truncated the observation errors so that the CPSR and QOR
experiments produced closer agreement with the assimilated and independent observations. This does not mean that the
assimilation of raw retrievals is incorrect. It means only that the reported observations errors may be too large because they
account for errors associated with the retrieval prior and consequently they require a longer study period to show an
assimilation impact compared to CPSRs.
Comparison of the CPSR experiments with IASI CO retrievals and MOZAIC *in situ* CO observations generally showed
improved agreement in the middle and lower troposphere compared to the MET experiment. For the IASI comparison, the
improvements were significant and extended from 250 hPa to the surface. For the MOZAIC comparison, two (Dallas, TX and
Portland, OR) of the three (no improvement for Philadelphia, PA) urban areas studied showed improvements between 500 hPa
and 800 hPa. Below 800 hPa, there was little to no improvement. Although the assimilation impacts when compared to
MOZAIC were not significant, the lack of a near-surface improvement was unexpected. However, the DOFS analysis in the
discussion of Figs. 3 and 4 showed that there were no near-surface impacts because after accounting for the observation error
covariance, the transformed averaging kernel had very little sensitivity to the near-surface CO. Other researchers have not
found that result because they have not accounted for the observation error correlations.
Comparison of the CPSR experiment with IASI and MOZAIC showed degraded skill in the upper troposphere (above 250 hPa
for IASI and above 500 hPa for MOZAIC) compared to the MET experiment. That degradation was significant for IASI but
not MOZAIC. It was attributed to the assimilation of biased retrievals above 300 hPa illustrating the need to extend the CPSR
method to truncated retrieval profiles. Section 4.3 explained the extension, and Section 5.2 compared the L10VMRR-RJ3
(assimilation of truncated raw retrieval profiles) and CPSR-RJ3 (assimilation of truncated phase space retrieval profiles)
experiments where we did not assimilate the biased MOPITT CO retrievals above 300 hPa. That comparison showed that the
L10VMRR-RJ3 and CPSR-RJ3 experiments produced similar results confirming the applicability of the CPSR approach to
truncated retrieval profiles. However, they also highlighted an important characteristic of assimilating truncated retrieval
profiles. Excluding the assimilation of some elements of the observation profiles can significantly alter the: (i) information
content of the assimilated observations; and (ii) the amplitude of the averaging kernel sensitivities. Those modifications can
combine to reduce the state variable correlations and the corresponding analysis increments. We are researching modification
of the CPSR extension to truncated retrieval profiles to address the reduced impact from not assimilating retrievals from
selected levels.
**Code and Data Availability**
The current versions of the WRF-Chem, WRF, WRFVAR, and WPS codes are available from the WRF download site at
http://www2.mmm.ucar.edu/wrf/users/download/get_sources.html. The current version of the DART code is at available at
https://www.image.ucar.edu/DAReS/DART/DART2_Starting.php#download, and the current version of the WRF-
Chem/DART branch is available at https://www.image.ucar.edu/DAReS/DART/DART2_Starting.php#download. The WRF-
Chem/DART branch is the same as the DART code except for inclusion of the WRF-Chem/DART system. There is no need
to down load both codes. Presently, there is no users guide available for WRF-Chem/DART. However, the authors have
prepared a slide presentation that describes much of the chemical data assimilation script function, variables, and organization.
Interested readers should contact the first author for a copy of that presentation and assistance with using WRF-Chem/DART.
The large-scale model's forecast and observational data used to run the ensemble forecast/data assimilation cycling
experiments described in the paper are generally available from the respective data distribution sites. That data set has not
been posted to a public site due to its size but is available from the first author upon request.
**Acknowledgements**
NCAR is sponsored by the National Science Foundation (NSF). Any opinions, findings, and conclusions or recommendations
expressed in this publication are those of the authors and do not necessarily reflect the view of NSF. This research is also
sponsored by National Aeronautics and Space Administration (NASA) grant NNX11AI51G. We gratefully acknowledge Chris
Snyder and Avellino Arellano for discussions that led to a better understanding of the transformed averaging kernels associated
with truncated retrieval profiles. We also acknowledge Louisa Emmons and Benjamin Gaubert for their thoughtful reviews
and helpful suggestions both of which improved the quality of this manuscript. We also acknowledge the use of data products
from MOPITT, IASI, and MOZAIC/IAGOS programs.

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

| Experiment | Assimilate meteorology observations | Assimilate MOPITT CO raw retrievals | Assimilate MOPITT CO CPSRs | Assimilate MOPITT CO QORs | Assimilate retrieval full profiles | Assimilate truncated retrieval profiles |
|---|---|---|---|---|---|---|
| MET | Yes | No | No | No | No | No |
| VMRR | Yes | Yes | No | No | Yes | No |
| L10VMRR | Yes | Yes | No | No | Yes | No |
| CPSR | Yes | No | Yes | No | Yes | No |
| QOR | Yes | No | No | Yes | Yes | No |
| L10VMRR-RJ3 | Yes | Yes | No | No | No | Yes |
| CPSR-RJ3 | Yes | No | Yes | No | No | Yes |

4    Table 1. Summary of the WRF-Chem/DART Forecast/Data Assimilation Experiments.

|  | CompAK 1 | CompAK 2 | CompAK 3 | Trace |
|---|---|---|---|---|
| **Full Histogram** | .9638 | .4785 | .0099 | 1.452 |
| **0.9 ≤ DOFS ≤ 1.1** | .8997 | .1174 | .0006 | 1.018 |
| **1.4 ≤ DOFS ≤ 1.6** | .9771 | .5188 | .0059 | 1.502 |
| **1.9 ≤ DOFS ≤ 2.1** | 1.016 | .8899 | .0518 | 1.957 |

Table 2. Average information content for each mode of the averaging kernel for the entire study period. CompAK 1 denotes
the average information in mode 1, CompAK 2 is for mode 2, and so forth. Trace denotes the total information content. "Full
Histogram" means all retrievals were considered. "DOFS" denotes the degree of freedom for the signal, and the different
DOFS rows identify the average information content for the different DOFS ranges and averaging kernel modes.

|  | CompAK 1 | CompAK 2 | CompAK 3 | Trace |
|---|---|---|---|---|
| **Full Profile** | .9638 | .4785 | .0099 | 1.452 |
| **Reject Top Three** | .7983 | .2851 | .0045 | 1.088 |
| **Reject Middle Three** | .7254 | .3849 | .0078 | 1.118 |
| **Reject Bottom Three** | .9335 | .3770 | .0065 | 1.317 |

Table 3. Average total and fractional information content for each mode of the averaging kernel for the entire study period.
CompAK 1 denotes the average fractional information in mode 1, CompAK 2 is for mode 2, and so forth. Trace denotes the
total information content. "Full Profile" means all retrievals were assimilated (i.e., none were discarded). "Reject Top Three"
means that retrievals at pressure levels < 300 hPa were discarded. "Reject Middle Three" means that retrievals between 300 hPa
and 600 hPa were discarded. "Reject Bottom Three" means that retrievals below 700 hPa were discarded.

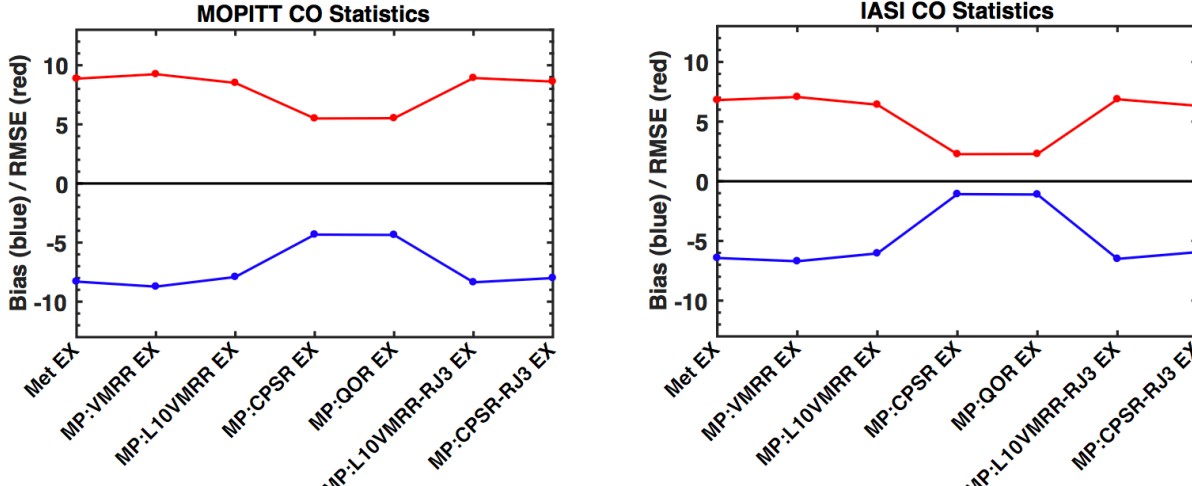

4   Figure 1. Forecast (assimilation prior) verification statistics for all experiments in MOPITT retrieval space on the left and

5   IASI retrieval space on the right. The red curve is root mean square error (RMSE), and the blue curve is bias (model –

6   observation). The experiments are described in the text and summarized in Table 1.

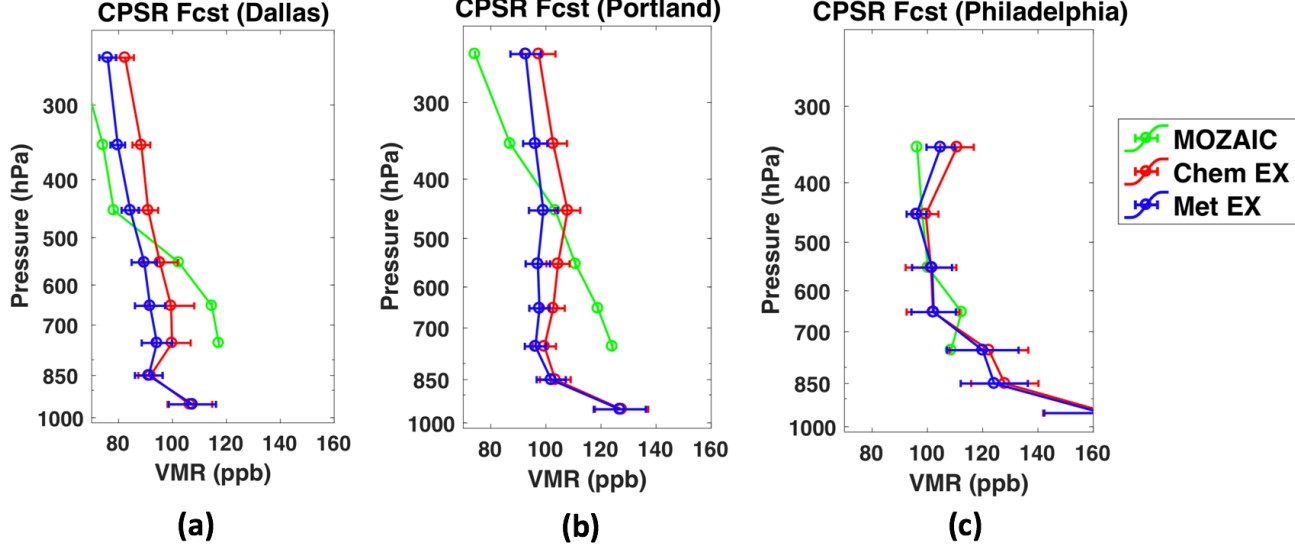

Figure 2. Comparisons of the CPSR experiment against the IAGOS/MOZAIC in situ CO profiles in ppb composited for 1 June 2008 for Dallas, TX in panel (a), 3 and 9 June 2008 for Portland, OR in panel (b), and 7 June 2008 for Philadelphia, PA in panel (c). Chem EX refers to the CPSR experiment. The error bars are based on the ensemble variability.

# MOPITT DOFS HISTOGRAMS

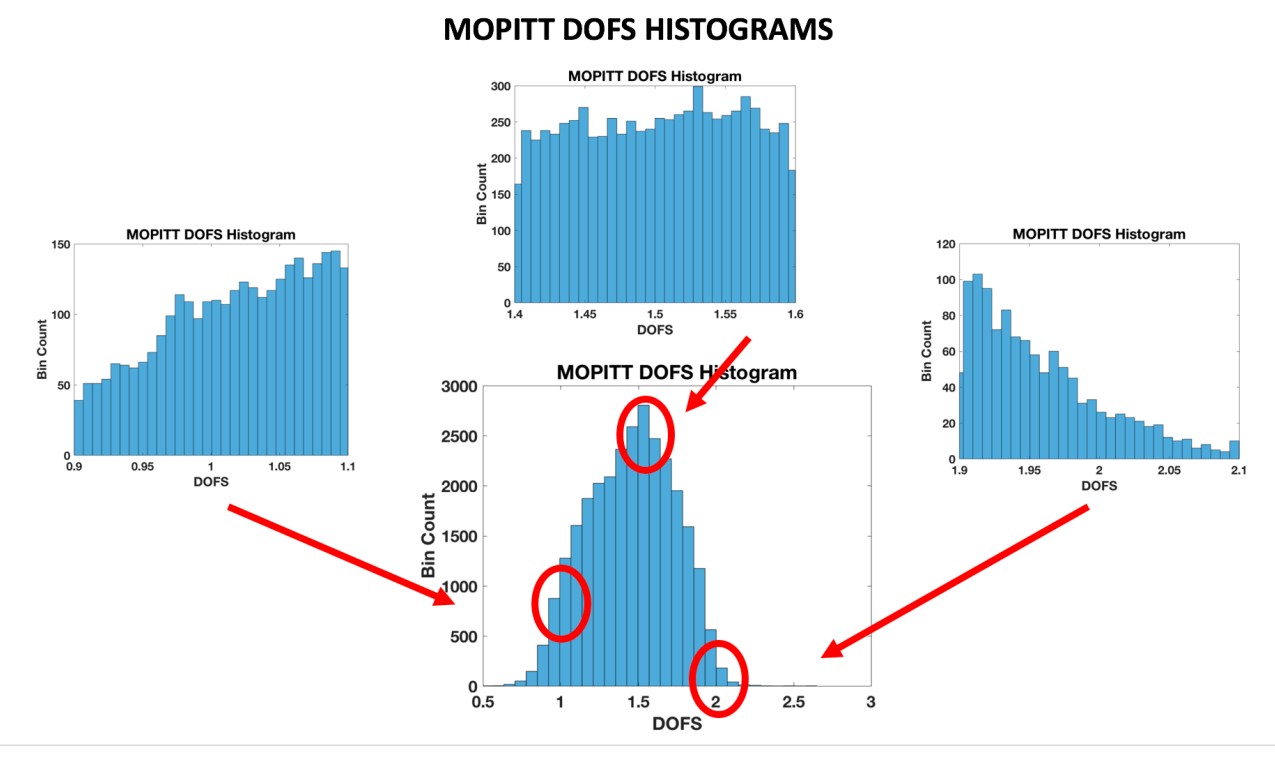

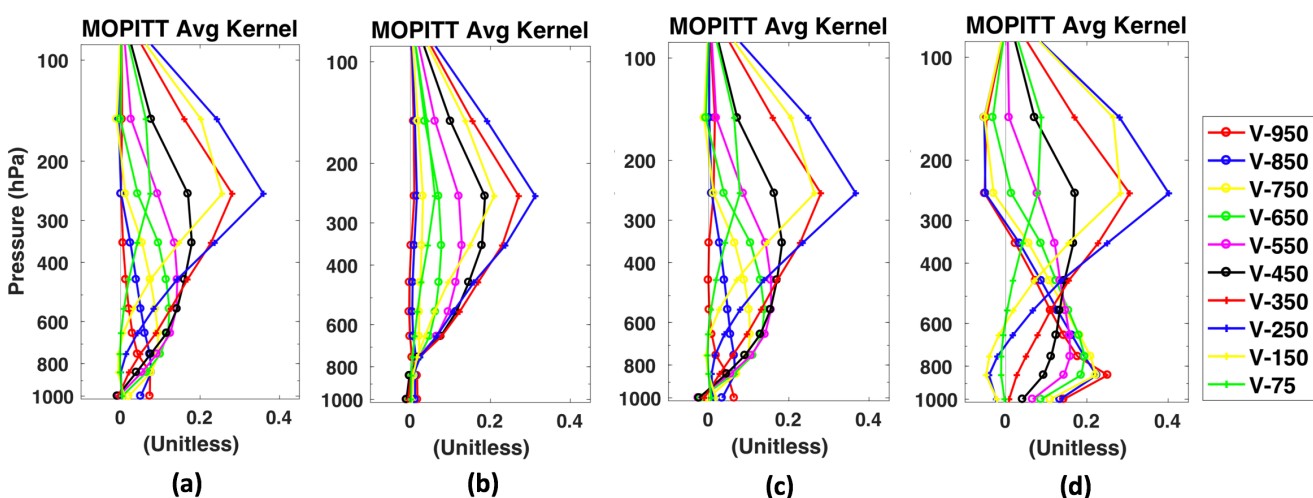

Figure 3. Histogram of MOPITT CO "degrees of freedom of signal" (DOFS) with blow-up histograms for selected DOFS ranges in the upper panels. The lower panels show composite MOPITT CO averaging kernel profiles for: (a) all DOFS, (b) $(0.9 \leq DOFS \leq 1.1)$, (c) $(1.4 \leq DOFS \leq 1.6)$, and (d) $(1.9 \leq DOFS \leq 2.1)$. The averaging kernel identifiers are V-xxx where xxx is the approximate pressure level mid-point in hPa for the associated averaging kernel profile.

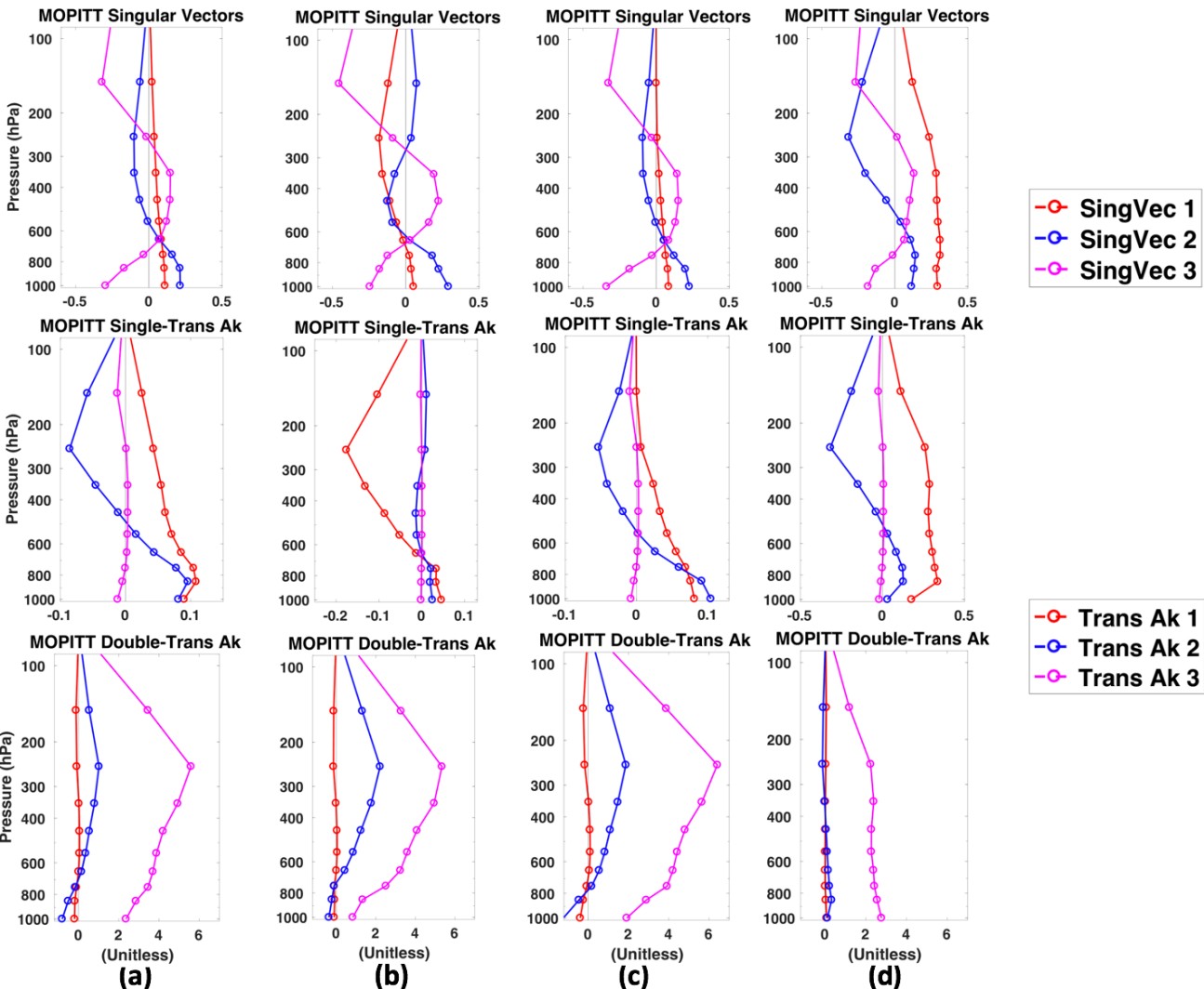

Figure 4. Composite vertical profiles for the: (i) leading left singular vectors of the MOPITT CO averaging kernels in the upper row, (ii) compressed averaging kernels in the middle row, and (iii) rotated and compressed averaging kernels in the lower row. The DOFS ranges are the same as defined for Fig. 2. For the profile labels "SingVec x" refers to ranked singular vectors where x = 1 is the first leading singular vectors, x = 2 is the second leading singular vector, and so forth. "Trans Ak x" refers to the compressed or rotated and compressed averaging kernel profile associated with the QOR and CPSR mode x respectively.

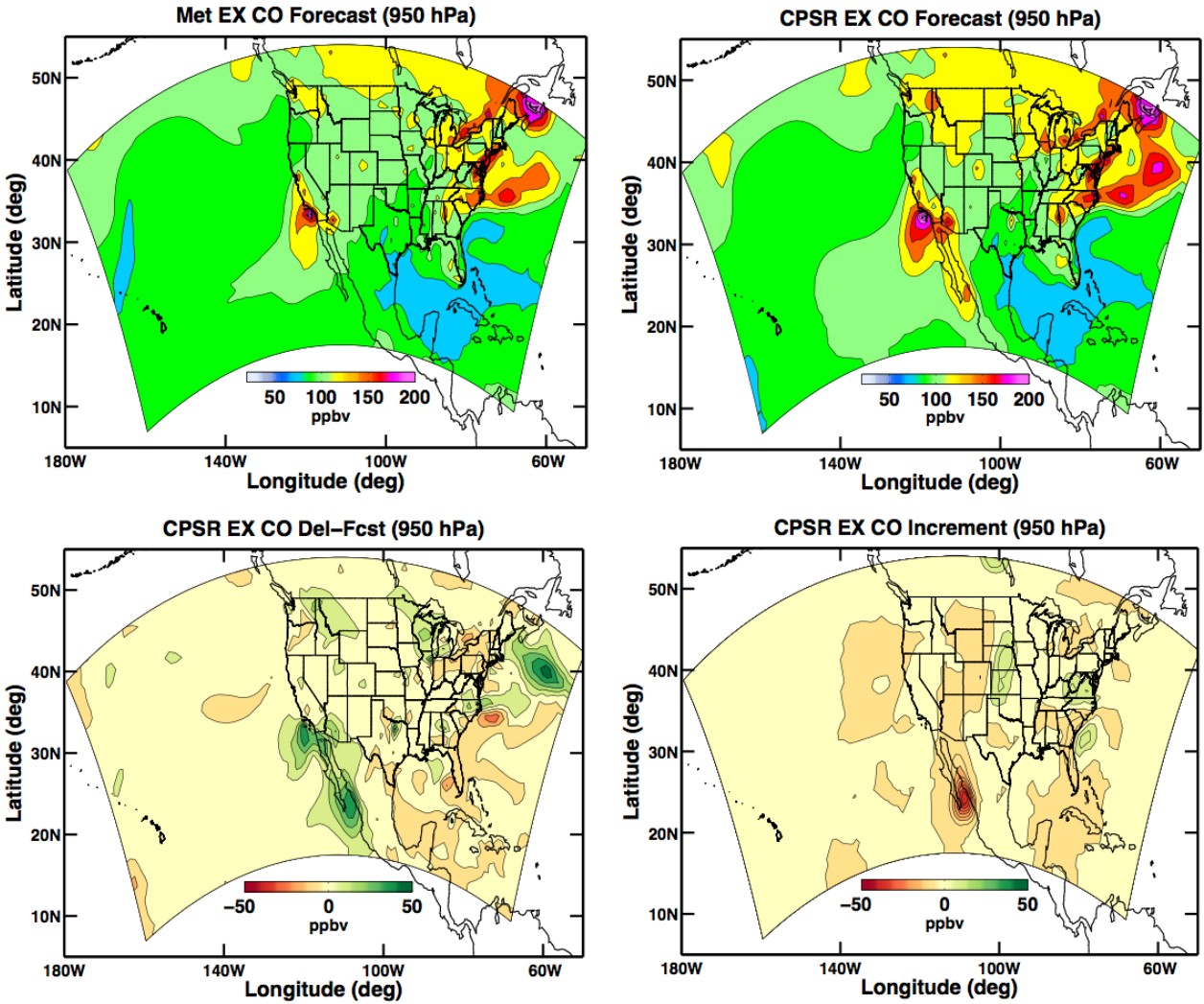

Figure 5. Shaded contours of CO in ppb for the MET and CPSR experiment 6-hr forecasts valid at this cycle time in the left
and right upper panels respectively. The lower row presents the difference between the CPSR and MET forecasts (the CPSR
experiment 6-hr forecast minus the MET experiment 6-hr forecast) in the left panel and the assimilation increment for analysis
at this cycle time in the right panel. All figures are for ~950 hPa and the 9 June 2008 18 UTC cycle. The curved rectangle
represents the WRF-Chem domain.

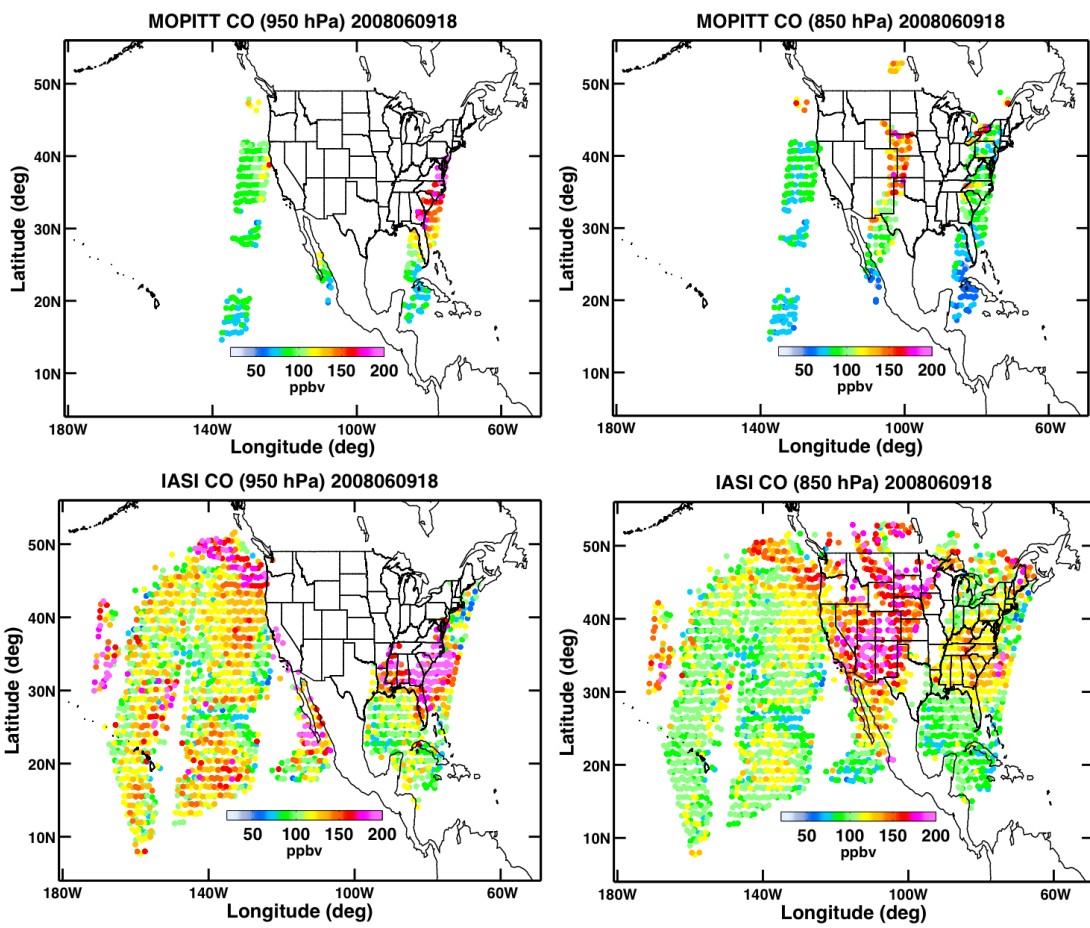

Figure 6. The assimilated MOPITT CO retrievals in the upper panels and the corresponding IASI CO retrievals (not assimilated) in the lower panels. The left figures are for ~950 hPa, and the right figures are for ~850 hPa. All figures are for the 9 June 2008 18 UTC cycle. The retrievals are in ppb.

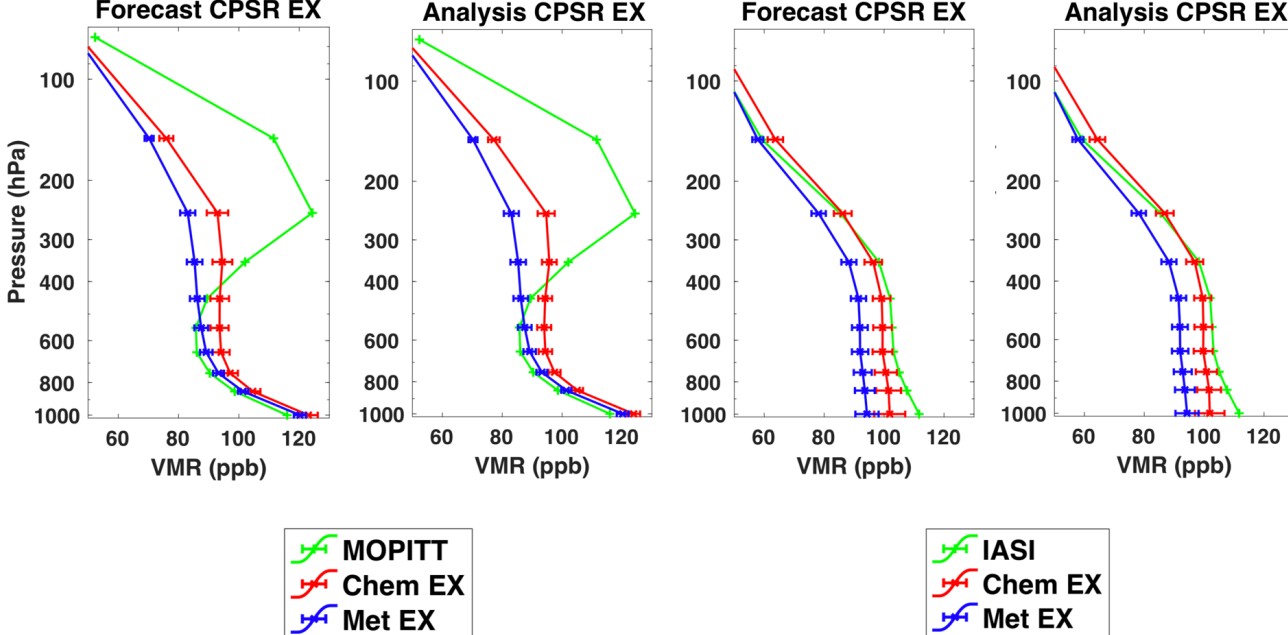

Figure 7. Vertical profiles of the time/horizontal domain average CO in ppb from the CPSR and MET experiments for 9 June 2008 18 UTC in retrieval space. "Forecast" is the assimilation prior, and "Analysis" is the assimilation posterior. The left two panels compare the forecast/assimilation results against MOPITT CO retrievals (assimilated), and the right two panels compare those results against IASI CO retrievals (not assimilated). In the legends, Chem EX refers to the CPSR experiment. The error bars are based on the ensemble variability.

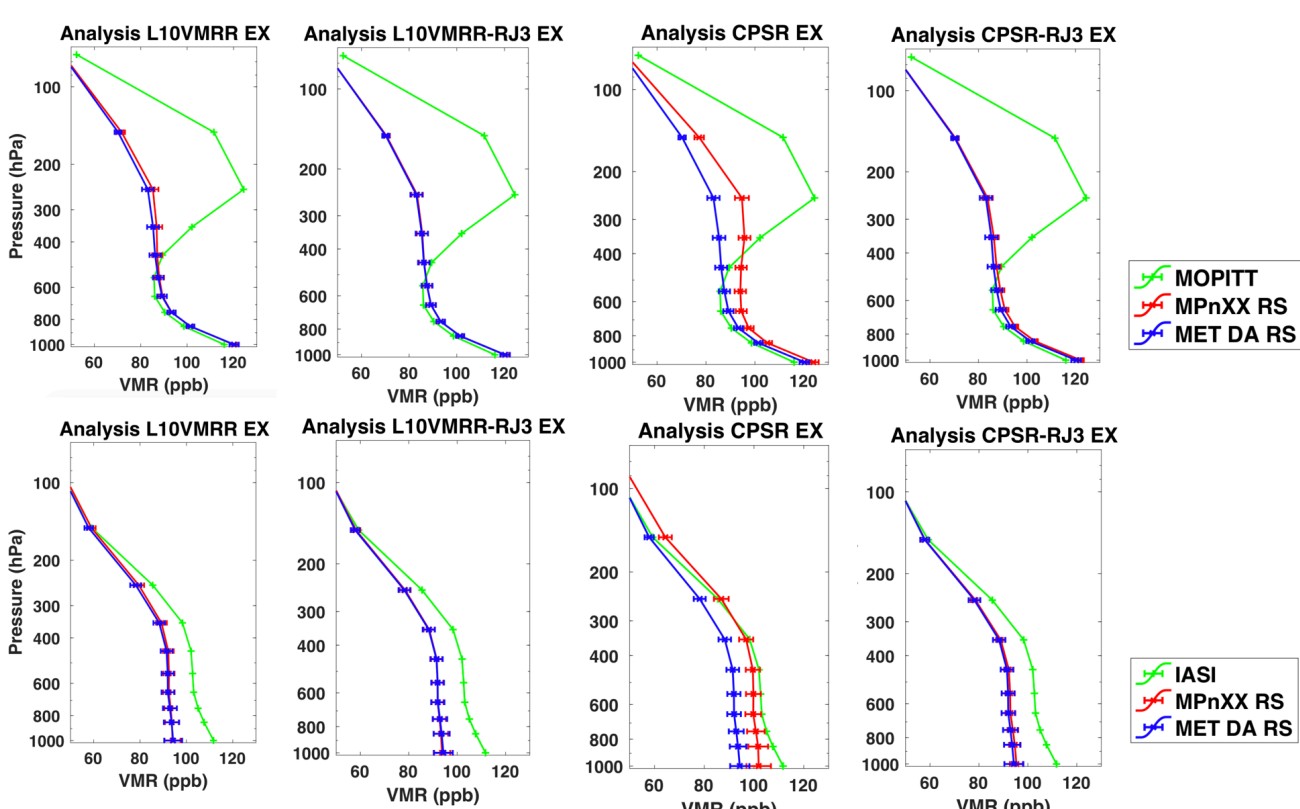

Figure 8. Same as Fig. 7 except this figure compares the L10VMRR, L10VMRR-RJ3, CPSR, and CPSR-RJ3 experiments. The upper panels compare the forecast/assimilation results against MOPITT CO retrievals (assimilated) and the lower panels compare those results against IASI CO retrievals (not assimilated). In the legends, Chem EX is a placeholder for the L10VMRR-RJ3, L10VMRR, CPSR, and CPSR-RJ3 experiments depending on the panel.

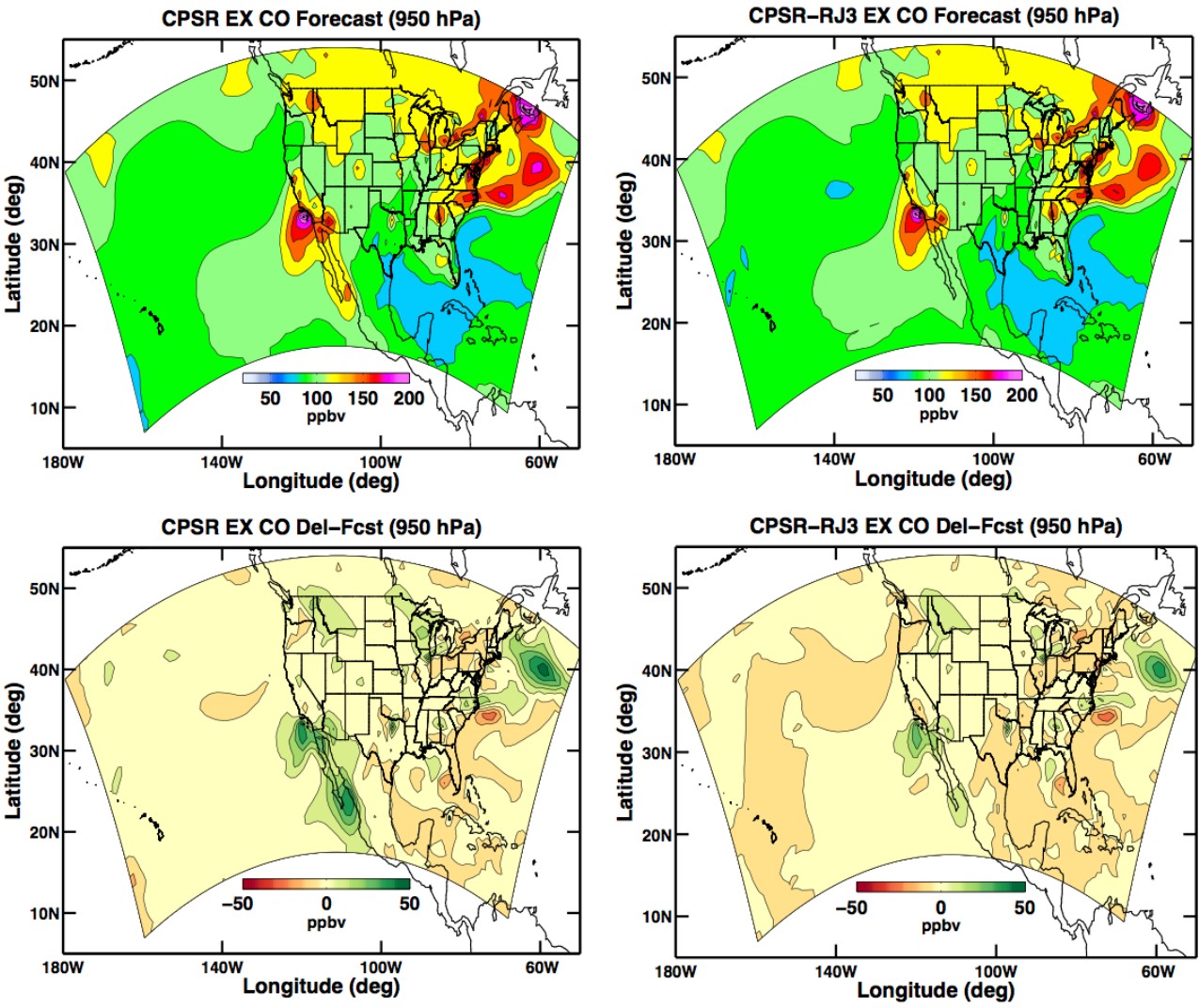

Figure 9. Shaded contours of CO in ppb for the CPSR and CPSR-RJ3 experiment assimilation priors in the left and right upper
panels respectively and for the CPSR and MET experiment difference (the CPSR minus the MET experiment, defined as CPSR
EX CO Del-Fcst) and the CPSR-RJ3 and MET experiment difference (the CPSR-RJ3 minus the MET experiment, defined as
CPSR-RJ3 EX CO Del-Fcst) assimilation priors in the left and right lower panels respectively. The CPSR experiments maps
in this figure are the same as in Fig. 5 and included for reference. All figures are for ~950 hPA at 9 June 2008 18 UTC.

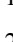

4    Figure 10. Same as Fig. 9 except for ~300 hPa.

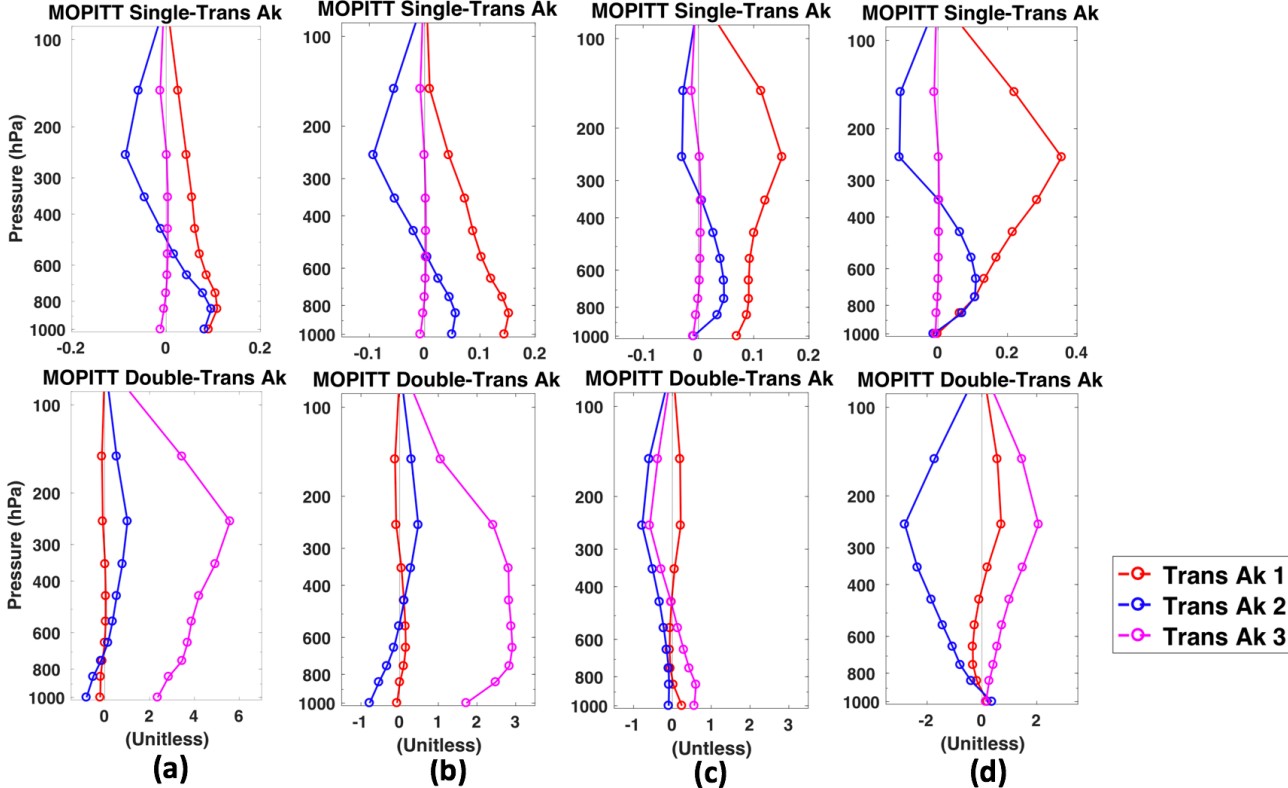

(a)  (b)  (c)  (d)

Figure 11. Same as the lower two rows of Fig. 4 except that this figure is for the retrieval discard experiments. Column (a) is
for the full retrieval profile assimilation experiment and is the same as column (a) in Fig. 3. Column (b) is for the "Reject Top
Three" experiment in Table 2. Column (c) is for the "Reject Middle Three" experiment.  Column (d) is for the "Reject Bottom
Three" experiment. Notice that the range of the abscissa is reduced from column (a) to columns (b) – (d).
