# Peer review of "Assimilating Compact Phase Space Retrievals (CPSRs): Comparison"

_Geoscientific Model Development, 2018_

## Referee Comment (RC1) · Anonymous Referee #1 · 8 May 2018

This study investigates the application of different methods for compressing the information content of satellite retrievals to the assimilation of CO from MOPITT. Results are compared to independent CO measurements from MOSAIC and IASI, overall showing a variable skill of the assimilation depending on elevation, pointing to a potential bias in the MOPITT data. A method is proposed to account for this bias, which is tested in the same framework and compared to results obtained without bias correction. The methods that are presented are interesting, although partially not new since this work

builds on earlier published work. The new part concerns a validation against independent data as well as the method to account for measurement biases. In my opinion, these two aspects are important to make a significant contribution beyond Mizzi et al (2016), which is needed for this work to be publishable. As explained further below, however, issues remain with both – as well as other conclusions that are drawn from this work, which remain to be addressed and call for significant revisions.

GENERAL COMMENTS

The assimilation period chosen for this work is only 9 days. Given the size of the domain, this is probably barely enough to move air from one side to the other side of the domain. This raises questions about the influence of the initial and boundary conditions that are applied. A large fraction of the RMS in Figure 1 is explained by a bias of the model relative to MOPITT. This is attributed to model errors in the first part but later to a MOPITT retrieval bias, which works rather confusing but more about that later. The trouble, which should be addressed in my opinion, is that parts of the domain are constrained by data whereas others or not (yet). The outcome of a validation comparison is then very dependent on whether the independent data are in the constrained or unconstrained parts of the domain. The Moziac comparisons include flights at June 3th and June 4th, i.e. the first part of the 9 days assimilation window. It should be made clear whether they are in data constrained regions of the domain of not, and whether the validity of the validation is compromised by the short spin up period. The same applies to IASI comparison. I was looking for information about co-location of IASI and MOPITT retrievals that were used but didn't find any. A more careful treatment of possible impacts of assimilation spin up is needed.

Looking at the validation results in Figure 1, 6, and 7, I'm not sure what to conclude since there are no error bars on the data nor the model estimates. Quite some discussion is spent on differences between the performance of MET, VMRR, and L10VMRR, in Figure 1. However, I doubt that any of that is significant. The same for the difference between CPSR and QOR. If I understand well, the MOZIAC profile is an average of

4 profiles. However, since the take-off and landing legs are in different cities, I suspect that the individual profiles could look rather different. In the case, I wonder how robust the altitude dependent under/overestimation really is (which is anyway hard to assess using only 4 profiles). To summarize, the statistics of the validation comparison requires more work.

On aspect of the retrieval compression I find very confusion, which is the information content before/after compression. I would think that you'd always loose information by filtering for dominant eigenvectors (whether in SVD or not). However, the text mentions several times that the uncompressed retrieval work less well because of larger observational uncertainty. I understand that the strategy is to select the most significant components of the averaging kernel, which are probably associated with the least uncertainty. However, those are contained in the unfiltered version of the assimilation also. Nevertheless, I also understand that if the poorly constrained directions introduce noise, this would deteriorate the assimilation performance. However, the evaluation has little to do with noise, but rather with bias. Somehow, the CSPR and QOR retrievals provide a stronger data constraint than their VMRR counterpart, pulling the solution stronger to the data reducing bias. However, it remains unclear to me how this happens, since preconditioning cannot add observational constraints. Could it be in the localization or some other part of the assimilation of the data?

A new element compared to Mizzi et al, 2016 is also the use of MOPITT super observations. However, it remains unclear how those observations are constructed. Issues are the averaging of data with different vertical sensitivities, averaging of averaging kernels and retrievals uncertainties. Given the importance of the averaging kernel and the retrieval error covariance for the method of compressing the data, further discussion is needed of how those are derived.

At first, I understood the validation in Figure 1 as the logical consequence of a bias in the model, explaining why both the assimilation and independent data statistics show similar improvements. However, the logic falls apart after a bias in the MOPITT retrieval

turns out to be the cause. In that case, you expect the assimilation methods that give the best agreement with MOPITT (in terms of bias) to give the worse agreement with IASI and vice versa. Why is the RJ3 experiment that is meant to account for the MOPITT bias performing poorer against IASI (Fig. 1, top right) than without this correction? What does it mean that CPSR-RJ3 performs better against MOPITT than L10VMRR-RJ3? (Fig. 1, top left) The suggestion is made that CPSR-RJ3 performs better, but you might as well argue that it performs worse because it ends up agreeing better with the biased MOPITT retrievals. It is concluded that the CPSR and QOR experiments perform better against MOZAIC than MET. However, looking at Figure 1 (bottom) I don't see this. QOR and CPSR are biased high because of the biased MOPITT retrievals. This is all fine, but how then can it be concluded that QOR and CPSR outperform MET in comparison to MOZIAC (page 19, line 22). All these points need thorough clarification.

SPECIFIC COMMENTS

Introduction: It reads like a duplication of the abstract. Yet, the introduction has a very different purpose, including how the current research fits into the work that is being done by other researchers. Currently, it mostly explains the relation between this work and Mizzi et al, 2016. A wider context is needed to provide the reader with a more general background of this research.

page 5, line 8-14: How many MOZAIC profiles have been used for validation?

Page 5, line 18: which state variables would otherwise be influenced by the CO data?

Section 4.1: the difference between regular and L10 retrievals is explained, but a more explicit link should be made to with is done in experiment VMRR and L10VMRR. The section about QOR and CPSR should return to this discussion since the equations that are shown there suggest L10VMRR are used although this is nowhere mentioned.

Page 6, line 20: 'Another reason is to include pre-processing methods that enable us

to not assimilate selected retrievals' I don't quite understand what is meant here.

Section 4.2: It remains unclear from the description if any filtering of dominant eigenvectors is applied to QOR, and, if so, based on what criterion.

Page 9, line 2: 'rank of A is greater ... i.e., n - k >= q' But this assumes that the elements that are removed are in the nul space of A, which need not be the case (for example, if the purpose of leaving out layers is bias correction).

page 10, line 4: The right order is VMRR -> MET -> L10VMRR ...

page 10, line 8: 'We have investigated our results and concluded that they are correct ...', What was done?

page 10, line 9-23: The two 'explanations' that are given discuss why CPSR and QOR result may be similar or different, but they don't explain what is referred to as 'the discrepancy' in line 9.

page 10, line 17: You mean that the observational error covariance is still singular when transformed to the SVD space?

page 11, line 13: A more quantitative discussion is needed here of the bias that is found, versus what has been reported MOPITT v5.

page 12, line 15: How about the opposite: MOPITT's lower tropospheric sensitivities influencing the upper troposphere through Ak smoothing?

page 13, line 15: 'Those results suggest ... most sensitive to CO in the lower troposphere' Why is that? The text above describes, but doesn't explain anything.

page 14, line 20 - 21: I don't really see this in the Figure.

page 16, line 8: L10VMRR-RJ3 does account for the observation error covariance, but you just don't diagonalize / reduce its rank, right?

page 17, line 12: 'The CPSR-RJ3 experiment skill improvement ...' How about the

significance of this?

page 17, line 21: 'Reject Top Three'. Besides the point that this explanation of the meaning of the "RJ3" experiment comes rather late, it is also not clear what justifies this method of MOPITT bias correction – given earlier publications about the nature of the bias.

Figure 4 - 5: The discussion in the text is hard to follow, because it requires comparing figure 4 and 5. Why not show model – data differences, before / after assimilation in one Figure?

Figure 10: Why do these plots show results for all levels, whereas they represent experiments in which retrieval results for certain levels are not taken into account. How can these levels nevertheless show up?

TECHNICAL DETAILS

Title: misses a closing parenthesis (maybe better to remove the details enclosed in parenthesis anyway)

page 9, line 3: remove 'changes in'

page 9, line 4: 'retrieval' i.o. 'observation'

page 12, line 15: 'artifact' i.o. 'artifice'

page 12, line 25 and onwards: 'sensitivity' io 'variability'

page 12. line 7: 'Fig. 3' i.o. 'Fig. 2'

page 16, line 4: 'Section V.C.'

Figure 4, title of the lower left panel: CPSR – MET

Figure 8, 9, titles of panels in the bottom row: What is 'Del-Fcst'?

[Figure]

2018.

---

## Referee Comment (RC2) · Anonymous Referee #2 · 16 May 2018

General Comments

This paper provides an evaluation of the impact of the CPSR approach for full and truncated MOPITT CO retrieval profiles as compared to the assimilation of the original retrievals in vmr and log(vmr). In this application, the applied quasi-optimal retrievals (QOR) include a transform to 'diagonalize' the retrieval error covariance matrix $E_m$. The CPSR approach is differentiated from the QOR by including the intermediate step

of applying a 'compression' transform to the averaging kernel matrix A.

The background information in the introduction is quite limited as if this paper was to be taken an extension of Mizzi et al. (2016). It would benefit from additional information. For example, the introduction could include mention of examining the level of influence in the vertical by the different approaches and provide related background on this aspect based on earlier work. The introduction could mention use of QOR in addition to raw retrievals in addition to CPSR in assimilation for evaluation against independent observations. It is important to mention of the benefits and or similarities of CPSR identified in Mizzi et al. (2016) specifically alluding to the assimilation not only of raw retrievals but also QOR, especially since QOR is applied in this paper in addition to raw retrievals and the paper wishes to further validate the CPSP approach. This is currently done in the abstract and could alternatively be done in the introduction. The introduction could bring up papers where MOPITT CO assimilation was performed and indicate related results – which might then be relevant in the results section.

The need for additional information would extend to information on the IASI CO (e.g. vertical resolution and range, average kernels, any information accuracy) and a bit more on MOZAIC CO (e.g. vertical range and resolution, amount of data used, accuracy, precision).

The evaluation is based on a very limited period. While this is not ideal, results do show a sensitivity to the observations. The comparison to observations in the paper can lack rigour though such as the rather qualitative visual comparison of Figs. 4 and 5 as oppose to an absent statistical comparison and evaluation from the fields and data of these figures. The representation of singular vectors in most figures without having a quantitative sense of their relative influence (i.e. resultant influence with the singular values) may be adding some potential ambiguity in assessing their relative importance. As well, there is an absence of accompanying statements and comparisons (with references) to MOPITT CO assimilation results from other papers.

Other than reduced computational time referred in Mizzi et al. (2016) regarding the CPSR approach (and the related advantage of 'compressing' the averaging kernel), the main take away message gathered by this referee regarding the benefits of CPSR and QOR with full profiles is in not requiring to introduce non-diagonal observation error covariance matrices (assuming averaging kernels are also used) in the assimilation as may/would be required to obtain similar results from the raw retrievals (with a priori removed), this pertaining to the vertical distribution of observation information in the analysis.

The aspect of removing biased observation elements through truncated retrieval profiles with the CPSR approach retains the computational efficiency (even though not quantified in this paper) but also notably reduces the influence of the MOPITT CO measurements in the lower troposphere. On the other hand, the comparison to IASI in Figure 1, suggests that not truncating the retrieval profiles would provide better overall results for CPSR and QOR even with the biased data. Might this be a reflection of how the results in Figure 1 are generated and the what they represent? Are there are any other papers which would have used truncated MOPITT CO profiles?

The comparison to MOZAIC above 400-500 hPa (Fig. 1) is presented independently from the comparisons to both IASI and MOPITT at these levels (Fig. 6). Exploiting the similarity of IASI and MOZAIC in comparison to MOPITT would need to be done directly, such as when discussing Fig. 6, as oppose to the reader needing to make this link in relation the MOPITT CO bias.

Another take away message and concern is the stated conclusion that assimilation of MOPITT CO raw retrievals shows little impact. This is attributed, in the paper, to having applied diagonal observation error covariance matrix (line 1 of page 10) if not also the increased observation error variances for the observation with removed a priori. The mention and consideration of other papers and accompanying results on MOPITT CO assimilation, such as Miyazaki et al. (2015) which show notable impact, would be necessary.

[Figure]

There is the tendency to very frequently use ':' followed by (i), (ii). ... It might be worth verifying if this can be reduced. As well, ':' is not really necessary in these cases.

While the paper reads fairly well, an overall revision is recommended to polish up the text.

Specific Comments

Abstract:

P1L18: The choice of 'results confirm' suggests that a computational assessment is performed and included in this paper, which is not the case. It may be a matter of rephrasing and or expanding, in the introduction, on the computational benefit indicated in Mizzi et al. (2016) in use of CPSR.

P1L23-24: Point (ii) is not specifically shown in this paper.

Introduction: (see also General Comments)

P2L10: This line is a summary line of a result of Section 5.1. Might best be removed by referring to issues and concerns to be addressed in the paper and not the results themselves.

P2L12: "In the second part of the paper" refers to what section? As well it assumes a first part which has not been specified explicitly (this referring the P2L10 above). It is suggested to begin this sentence (if kept) instead with 'Therefore, we ....'

P2L13: "The rest of this paper" would best be replaced by "This paper" considering P2L12 above and that the results section is also alluded to below.

P2L14-18: Sections 2, 3, 4, and 5 instead of II, III, IV and V. This applies to one or two more places in the paper.

P1L16: '... and an extension of CPSRs' (added 'an')

P1L17: Might be worthwhile to refer here to the content of the two subsections in

[Figure]

Section 5.

Section 2:

P3L9, P3L10, P4L3, : '... the DART' (added 'the')

P4L12: 'is much finer' instead of 'is much greater'

P4L20: 'Miyazaki' instead of 'Miyazki'

Section 3: (see also General Comments)

P4L25: 'In the first part of the paper' is actually intended to refer to the first part of the results section. Best to re-phrase.

P4L23 and P5L1: Suggestion - 'independent observations from the IASI instrument and the MOZAIC project.' or something similar. (best to remove parentheses)

Section 4:

P5L17: '...difference are: (i) the ...' or '...differences are (i) the ... and (ii) the ...' or ...

P5L17: There is also the number if days (9 days instead of one month).

P5L18: Does (iii) actually refer to univariate CO assimilation as oppose to localization – unless this is what is meant here by localization (i.e. not be being coupled to MET assimilation in this case)?

P5L24: It might be worthwhile to mention whether CPSR and QOR use vmr or log(vmr) for 'y' in the actual assimilation application (especially since A is applied/provided for log(vmr)). If log(vmr) then equation (2) and (3) would be good as is as long as 'y' is defined accordingly.

P6L1-6: Phrasing could be improved and simplified

P6L3: Change Section V.A

P6L16 and P6L18: The two lines referring to Gaussian/non-Gaussian distributed errors seem to contradict each other somewhat.

P6L20-21: Not clear on the value/meaning of this last sentence.

P6-7: Equation numbering not aligned (as would be from use of 'right-justified')

P6L18-19: Point (ii) could refer to Eq. (2) and QOR to make even clearer the relationship between QOR and CPSR.

P6L20-P7L1 and P7L4-P7L8 are somewhat repetitive. Maybe part P6L20-P7L1 could be removed with some changes for an introduction to what follows.

P8L20-24: As pointed out earlier, one could point to Eq. (2) and QOR for this part.

Section 5:

P9L13: How about the the MET and CO assimilation not being coupled (or being localized?) as per P5L18.

P9L14 and top of Figure 1: What are the units? Maybe unitless because both are referring to log(vmr)? Do these sum up the contributions from all vertical levels? Out of curiosity, how large are these values relative to the observation and background error standard deviations? This might be useful to compare with the RMSE.

P10L1: Due only to discarding the observation error cross-covariances and not also due (at least partly) in removing the a priori effect? Just wondering? A comparison to other papers also assimilating MOPITT CO might be pertinent here.

P10L4 and Figure 1: Would be better to split Fig. 1 in Fig. 1 (for upper panels) and Fig. 2 (for lower panels)

P10L4: Use of arrows might not be best.

P10L4-P10L23: Would some or much of this have been stated in Mizzi et al. (2016)? If so, might be best to reduce the text.

P10L25-P11L8: There is mention of the increased bias with MOZAIC from CPSR and QOR, this supported also by IASI CO in Fig. 6 and related to the MOPITT bias (also displayed by compared MOPITT and IASI in Fig. 6 – if IASI has comparatively no or less bias?)

P11L13-15: Any CO assimilation papers showing or not some impact near/at the surface?

P11L12: 'little or no change' instead of 'little or no improvement' as whether or not there is any improvement is not shown here.

P11L19: The Fig. 2 blow-up histograms are not really needed. It's up to the authors. Might it be best to split the histogram and the lower panels into two separate figures?

P12L1-4 (and beyond): Could differences in the vertical of the CO background (forecast) error variances/covariances also be a contributing factor to some degree, this depending on the assimilation setup? Having some sense of the variation in the vertical of error variances (and error correlations) might be beneficial. Would differences in background error covariances in different papers contribute to explaining differences in results?

P12L12: Was any scaling really needed?

P12L7-22 (and beyond): See General Comments on the display of the singular vectors.

P12-P13: I only skimmed the text for the review on these pages.

P13L17-18: One might question the application of the scaling in the first place.

P13L18: e.g. '. . . that, when . . . is considered, the . . .' (while this is likely somewhat subjective, adding some commas here and or similarly elsewhere in the paper might be considered)

P13L20: '. . . and the first . . .' (added 'the')

[Figure]

P13L22: Might the validity of this assertion depend on the singular values?

P14L2-3: Please indicate actual references and elaborate on results where applicable.

P14L3: What is meant by 'do not adjust for the averaging kernel linear dependencies or for the observation error covariance"s'". [Might the latter be in reference to not including error correlations (cross-covariances)?]

P14L11-13 (and remainder of the paragraph): While there is some level of consistency in the coastal regions, it is not that evident that one could say that the analysis and forecasts are 'generally consistent' with the observation. Maybe some re-phrasing would be needed. A quantitative evaluation might help.

P14L15-16: Has (i) been looked at to some degree?

P14L16:17: Has (ii) been verified?

P14L17: The changes in the analyses seem rather weak in the central U.S. or there-abouts in comparison what is needed to increase the analysis to levels fairly close to what is seen in Fig. 5. Might a quantitative evaluation help?

P14L20: Does this refer to the central U.S. or is an overall assertion? It is not so clear from the figures if for the central U.S.. Either way, a quantitative evaluation (by regions maybe) might be more meaningful to justify this assertion (and those above).

P14L25: Might it be worth to mention/discuss the level of similarity and differences between 'SS' and 'RS' profiles?

P15L3-6: 'for pressures less than about 500 hPa, the MOPITT CO assimilation with CPSR draws the forecast and analysis further away from IASI while the opposite occurs for larger pressures.'

P15L3-6: Could refer to the comparison to MOZAIC in Fig. 1 to support the comparison with IASI in the upper levels.

[Figure]

P15L11-21: An alternative would be for a version of this 'summary' to instead be in the 'Summary and Conclusions' section. It's up to the authors.

P15L14: It is not really that the 'phase space' observations error variances is reduced as oppose to the transformation allowing to account for the otherwise neglected 'retrieval space' error correlations.

P15L16-17: Part of (ii) is actually a repetition of (i). Some change in the sentence is needed.

P15L17: As part of (ii), has the statement 'linearly dependent portion of the transformed retrievals do not . . .' (repeated earlier as well) been verified, noting that background error covariances (and its non-zero error correlation coefficients) contribute to determining the distribution of information for strongly overlapping averaging kernels (likely requiring more computational effort though). Any other references for his part (e.g. Migliorini, 2008 and or 2012 or even Mizzi 2016)? If so, they should also be indicated earlier on in this paper.

P15L21-23: Have other assimilation studies shown this as well – that the resulting CO analyses and forecasts in the upper levels would be biased. This result would be expected considering the literature on the MOPITT CO data – assuming IASI and also MOZAIC CO is less biased. Might be good to indicate that this was not entirely un expected.

P15L23. This also applies to the comparison with MOZAIC CO.

P16L4: Section V.C to be changed.

P16L8: '. . .accounts for the error correlations of the observation error covariance matrix.'

P16L11: e.g. 'that, in the upper troposphere, the' (commas)

P16L17: e.g. 'troposphere, there'

P16L18: 'A comparison with'

P16L21-22: Could be re-phrased.

P17L1: Remove 'However', i.e., 'The forecast . . .'

P17L3: '. . . United States similar to, though weaker than, the CPSR experiment' or something similar

P17L5-7: e.g., 'The upper tropospheric impacts of Fig. 9 show even smaller changes for the CPSR-RJ3 experiment except for the reductions over the southeastern United States. The CPSR-RJ3 experiment therefore further demonstrates, in addition to Fig. 7, the reduction of bias in the upper troposphere through the removal of the biased observation profile elements, this though at the expense of reduced improvements in the lower troposphere.'

P17L9-11; This should explicitly refer to the upper right-hand side panel with the comparison to IASI CO.

P17L11: The improvement is rather small though as compared to CPSR (and QOR) in Fig. 1. This needs to be indicated. Is this related to how this diagnostic is generated, e.g. maybe because of a dominance of the lower tropospheric RMSE contributions (as compared to the upper layers)?

P18L5-6: This should refer to the levels with pressures below about 500 hPa. 'Significantly' seems to be an exaggeration based on the curve. I suggest removing 'significantly'.

P18L7-13 and Figure 10 (with Table 2): The bottom row of Fig. 10 (even in combination to Table 2) suggests that the 'reject middle three' may have least impact in assimilation. This would be contrary to just looking at the traces in Table 2 which, based on the earlier statement, indicate the 'reject bottom three' provide the least amount of info. Am I missing something? Any discussion or comments.

P19L9: Removing 'likely' seems appropriate as it seems pretty certain.

Summary and Conclusions:

P19L16: Instead of 'magnitude of the observation errors' is it more the omission of the 'observation error correlations in the assimilation' in comparison the CPSR and QOR effects?

P19L19: 'Truncated the observation errors' may not be the correct wording considering the above.

P19L22-23: Applies also to IASI CO. Even better would be to instead mention MOZAIC CO at P20L5.

P20L2: 'because, by accounting for . . . error correlations ,'

P20L21-P21L3: Different contradictory statements in this sentence related to the impact at the surface. Also, one could mention the approximate proportion of cases where surface impact may occur. One might also consider the background error variances in also contributing to the level of impact at near the surface (on top of the averaging kernels themselves (and obs error covariances))

P20L10: 'confirming the applicability of the CPSR . . .'

P20L13: 'Excluding the assimilation of some elements of the observation profiles can . . .'

P20L16: 'to address the reduced impact from not assimilating retrieval profile levels' ('reduced' instead of 'remote' and . . .)

Additional remarks on Tables and Figures:

Table 1: Might be better to follow the form of Table 1 in Mizzi et al. (2016)

Figures: Font sizes for panels with y-axis as pressure are on the edge of being too small or are too small. Please check.

Figures 6 and 7: For clarity, might be best to drop the 'SS' results (at least for Fig. 7 if not both). That is unless the one intends to mention and discuss in the text, for Fig. 6 for example, the level of similarity and differences between 'SS' and 'RS'.

Figure 10: 'except that this figure' (added 'that') Figure 7 (lower panels): Unless this is a visual clarity issue, it seems that the Met EX RS results near the surface differ between the CPSR panels and the L10VMMR panels, while they would be expected to be the same. Please check.

References: I did not check the format of the references.

---

## Author Comment (AC1) · 10 Aug 2018

RESPONSE TO REVIEWER 1 General Comments: 1. Domain Size: We agree that the study period is such that air barely moves from one side of the domain to the other. But it gets most (if not all) of the way across, so the study period is sufficient to determine whether the assimilation can address errors introduced by the initial and/or boundary conditions. Additionally, we are concerned with the grid interior, and the air

easily crosses the interior. 2. Bias: We are concerned with three types of bias for the chemistry: (i) bias introduced by the initial and boundary conditions, (ii) bias introduced by the model, and (iii) bias introduced by assimilating bad observations. For the MET experiment the bias is due to initial/boundary condition and/or model errors. For the chemical assimilation experiments, the goal is to reduce the initial/boundary condition errors and thereby improve the forecast. Figure 1 shows that for the L10VMRR, CPSR, and QOR experiments, the assimilation of MOPITT CO retrievals generally reduced the bias and improved the model forecast skill. Despite that increased skill, when we compared the improved forecasts against independent data (the MOZAIC and IASI data) we found that the assimilation had introduced a bias in the upper troposphere. That bias is not due to initial/boundary condition error. Our analysis shows that it is due to the assimilation of biased observations. The improvements in Figure 1 when comparing against IASI are in spite of the bias introduced from assimilating these biased observations. 3. MOPITT Coverage: It is correct that for any particular assimilation cycle a portion of the domain is constrained by MOPITT observations and a portion is not. Such spatial and temporal sparsity is a characteristic of polar orbiting satellite observation platforms. Over time such platforms observe the entire globe. MOPITT observes the entire globe in four days. So that during our study period MOPITT observes the entire domain at least twice. 4. Spin-up Period: The results in Mizzi et al. (2016) suggest that for phase space assimilation experiments the spin-up period for stable verification statistics is two to three days. Our study period is long enough to account for that spin-up period. 5. MOZAIC Data: The MOZAIC data came from ascent and descent soundings on: June 1 and 5, 2008 at Dallas, TX; June 3 and 9, 2008 at Portland, OR; and; June 7, 2008 at Philadelphia, PA. That is ten profiles which cover our study period which ran from June 1 to 9, 2008. The June 1, 2008 profiles were taken during our spin-up period so they were discarded. The discussion of the MOZAIC comparison has been revised to address the reviewer's comments. See P12 L22 to P13 L18. 6. IASI Data: IASI has global coverage nearly every two days, and MOPITT has global coverage every three to four days. Therefore, there were times and locations where

the MOPITT and IASI observations were coincident. We agree that assimilation results for such coincident location are likely be better than those for non-coincident locations. We are interested in whether the assimilation improves model performance in a bulk sense and do not believe that the results from the coincident (or nearly coincident) locations dominate those bulk verification scores because the areas of overlap are small compared to the domain size. 7. Significance of Skill Differences for MET, VMRR, L10VMRR, CPSR, and QOR: The differences between MET and VMRR and for MET and L10VMRR are not significant. Those between MET and CPSR (or QOR) are likely to be significant. For the experiments that are common to those in Mizzi et al. (2016), we found that the magnitude of the differences is comparable to those found in Mizzi et al. (2016) who found the significance to be as described above based on applying the Student t test to the difference of two means. We revised the figures to include error bars and discussed this in captions and text. See Figs. 2, 7, and 8. 8. Information Content before and after Compression: When using a Singular Value Decomposition (SVD) to compress data, there is a difference between discarding modes with zero singular values and discarding modes whose singular values are non-zero but small relative to the leading singular values. Arguably, due to round-off error in a digital world it may be difficult to distinguish between the zero and small non-zero singular values. However, due to the nature of the retrieval process and because one of the reported products is "degrees of freedom of signal" (DOFS) – the trace, we know the number of non-zero singular values for an SVD of an averaging kernel with an accuracy of one because the DOFS is reported as a real number, and it is arbitrary whether to round the fractional part up or down. Now assuming that we know the number of non-zero singular values, we can perform the compression by discarding the zero singular values and associated singular vectors without loss of information. That is what is done in the compression step of the CPSR transform. If we had discarded modes whose singular values are non-zero but small, then there would have been information loss. 9. If the Information Content before and after Compression Is the Same, Why Do CPSR and QOR Produce Improved Skill? The CPSR experiment can have better verification

scores compared to VMRR and L10VMRR because the: (i) correlations are greater, and/or (ii) transformed observation errors are smaller. We think it is primarily due to smaller observation errors. They are smaller due to the compression step of the CPSR transform. They cannot be smaller due to the diagonalization step because that is a variance maximizing rotation. If the compression step had no filtering effect on the errors, then the variance resulting from the diagonalization step would be no smaller than that from the compression step. The QOR experiment produces the same results as the CPSR experiment because the linearly dependent portions of the averaging kernel do not contribute to the assimilation results (this point is discussed in greater detail below and revised in the text). See P11 L21 to P12 L23. 10. Construction of the Super-Observations: The paper states that the super-observations were constructed as follows, we: (i) sort the retrievals, retrieval priors, averaging kernels, and retrieval error covariances into bins that are ∼90 km square, (ii) calculate the binaverage for each of those variables, and (iii) assimilate the bin-average retrievals. We use an arithmetic average (as opposed to error covariance weighted average) when calculating the super-observation and do not apply a correction to the retrieval error covariance super-observation because we are interested in the impact of the reported errors and can apply an error tuning factor to adjust the errors and balance the observation fit as needed. Other studies e.g., Eskes et al. (2003), Miyazki et al. (2012 a and b, 2015), and Barre et al. (2016) have used similar super-observation strategies. We believe that the description of this process here and in the text of the revision is sufficient to describe our methodology. See P6 L7 to P6 L17. 11. Why Do the RJ3 Experiments Perform Worse against IASI? We answer this question in the discussion of Table 3 and Fig. 10. When one discards retrievals at a certain level (for discussion consider the top three levels as done in the paper), it impacts the amount of remaining information to be assimilated (Table 3 – the difference between the first and second row) and the sensitivity of the resulting averaging kernel (see Fig. 10 – the difference between columns (a) and (b). Those changes combine to remove most of the beneficial impacts from assimilating the MOPITT observations. In effect assimilating retrievals at MOPITT's

top three levels positively impacts the middle and lower troposphere through the averaging kernel. When those retrievals are not assimilated those positive impacts are not realized. See P19 L24 to P21 L14. 12. Why Do We Conclude that CPSR and QOR Perform Better against MOZAIC than MET? The paper has been revised to address this comment. See P12 L25 to P13 L21 and the revised Fig. 2. Specific Comments: 1. Introduction: It reads like a duplication of the abstract. Yet, the introduction has a very different purpose, including how the current research fits into the work that is being done by other researchers. Currently, it mostly explains the relation between this work and Mizzi et al, 2016. A wider context is needed to provide the reader with a more general background of this research. We have revised the introduction to provide a wider context as requested. See P1 L11 to P4 L9. 2. page 5, line 8-14: How many MOZAIC profiles have been used for validation? Eight profiles were used. This is discussed in Section 5.1 (page 10, lines 8-11). That seems like the appropriate place because this section is the general introduction of the observational data used for validation, and Section 5 discusses specific application of the validation data to this paper. 3. Page 5, line 18: which state variables would otherwise be influenced by the CO data? There are two types of correlations that this localization is concerned with: (i) correlations that have been observed in the field e.g., CO and O3 are known to be correlated, and (ii) correlations in the ensemble. In the ensemble Kalman filter, the update depends on the correlation in the ensemble. Some of those are real correlations i.e., they are observed in the field (or thought to be real based on the chemistry) and some are spurious. Historically, in chemical data assimilation, all correlations except those between the observed species and the corresponding state variable are turned off through localization. In this paper, we used the customary localization for chemical data assimilation. 4. Section 4.1: the difference between regular and L10 retrievals is explained, but a more explicit link should be made to with is done in experiment VMRR and L10VMRR. The section about QOR and CPSR should return to this discussion since the equations that are shown there suggest L10VMRR are used although this is nowhere mentioned. Revised, see P8 L7 to P8 L18. 5. Page 6, line 20: 'Another

reason is to include pre-processing methods that enable us to not assimilate selected retrievals' I don't quite understand what is meant here. Revised, see P8 L17 to P8 L18. 6. Section 4.2: It remains unclear from the description if any filtering of dominant eigen- vectors is applied to QOR, and, if so, based on what criterion. For the QOR and CPSR experiments there is no filtering of the dominant singular vectors (i.e., filtering those whose singular values are non-zero). One problem is identifying the non-zero singular values. The paper is revised to make this more clear. P9 L8 to P9 L9 and P10 L13. 7. Page 9, line 2: 'rank of A is greater ... i.e., n - k >= q' But this assumes that the elements that are removed are in the null space of A, which need not be the case (for example, if the purpose of leaving out layers is bias correction). This does not assume that the discarded elements are in the null space of A. The dimension of A is n and the rank of A is k. After discarding q elements of the retrieval profiles the dimension of the revised A (call it A) is n – q by n. The rank of A i.e., the number of non-zero singular vectors will be less than or equal to k. Generally, k « n so that discarding q elements of the retrieval profile has little or no impact on the rank of A i.e., as you observed the discarded elements of the retrieval profile are in the null space of A. If the discarded elements are not in that null space, then the rank of the truncated A is less than that of full A. The paper is revised accordingly. See P10 L19 to P11 L1. 8. page 10, line 4: The right order is VMRR -> MET -> L10VMRR ... Corrected. 9. page 10, line 8: 'We have investigated our results and concluded that they are correct ...', What was done? In preparation of this paper, we revised the computer code used by Mizzi et al. (2016). The revised code incorporates the QOR code into the CPSR code as opposed to using separate codes as was done in the earlier paper. When writing this paper, I had forgotten that we had similar results between CPSR and QOR in the earlier paper. So, it is true that I spent time double-checking the code etc. to confirm that these results were correct. During preparation of the revisions I reviewed the earlier paper. Therefore, this has been revised in this paper. See P12 L10 to P12 L22. 10. page 10, line 9-23: The two 'explanations' that are given discuss why CPSR and QOR result may be similar or different, but they don't explain what is referred to

as 'the discrepancy' in line 9. As discussed in the preceding comment, the paper has been revised because there is no discrepancy. However, this discussion of why the CPSR and QOR results are similar is still apt. The paper states that: [c]onsequently, the CPSR and QOR experiments yield similar results because: (i) the QOR experiment apportions the error and assimilates the linearly dependent modes (which have little or no impact), while (ii) the CPSR experiment apportions the error and does not assimilate the linearly dependent modes. 11. page 10, line 17: You mean that the observational error covariance is still singular when transformed to the SVD space? No, in the QOR experiment, the averaging kernel is rotated with the leading singular vectors from the observation error covariance matrix. The averaging kernel matrix is singular and so is the rotated averaging kernel matrix. Generally, for this work the observation error covariance matrix was not singular. Revised P9 L9 to P9 L10. 12. page 11, line 13: A more quantitative discussion is needed here of the bias that is found, versus what has been reported MOPITT v5. Deeter et al. (2013) reported a positive bias in the MO-PITT CO retrievals of ∼14% in the upper troposphere. Martinez-Alonso et al. (2014) suggested that the MOPITT CO retrievals were not biased. Based on those papers it is uncertain whether the retrievals but subsequent researchers e.g. Barre et al. (2016) have treated the MOPITT retrievals as biased in the upper troposphere. The results in Fig. 1 for the CPSR and QOR experiments suggest that the MOPITT retrievals in the upper troposphere are positively biased by at least 8% (likely more because the assimilation adjusts the analysis to lie between the prior and the assimilated observation). 13. page 12, line 15: How about the opposite: MOPITT's lower tropospheric sensitivities influencing the upper troposphere through Ak smoothing? Please provide additional explanation. We do not understand the comment in light of the cited text. 14. page 13, line 15: 'Those results suggest ... most sensitive to CO in the lower troposphere' Why is that? The text above describes, but doesn't explain anything. Deeter et al. (2007) report that MOPITT CO retrievals have sensitivity to (i.e., can observe) CO in the lower troposphere. Our results from assimilating MOPITT CO do not show improvement in the lower troposphere. The analysis of Figs. 2 and 3 shows why the assimilation results

do not have improvements in the lower troposphere. Fig. 2 confirms our conceptual understanding of MOPITT CO retrievals that the DOFS needs to be in the neighborhood of 2.0 to have sensitivity to CO in the lower troposphere. However, due to linear dependencies in the averaging kernel profiles that make up the composite profiles in Fig. 2, it is possible that sensitivities to the lower troposphere are masked for DOFS in the 1.0 and 1.5 figures. Figure 3 looks at the sensitivities for the linearly independent averaging kernel profiles. Figure 3 (second row) shows that for all DOFS categories, the linearly independent averaging kernel profiles have sensitivity near the surface. But when the error covariance is considered (Fig. 3 third row), that lower tropospheric sensitive disappears. Without that sensitivity, the assimilation cannot adjust CO in the lower troposphere. See discussion P14 L1 to P16 L14. 15. page 14, line 20 - 21: I don't really see this in the Figure. The text has been revised to make this more clear. P16 L16 to P17 L9. 16. page 16, line 8: L10VMRR-RJ3 does account for the observation error covariance, but you just don't diagonalize / reduce its rank, right? L10VMRR-RJ3 is the same as L10VMRR except that retrievals above 250 hPa are not assimilated. Neither of these experiments account for the observation error covariance. Diagonalization of the observation error covariance matrix is done by discarding the covariance terms (leaving only the error variance). This is the conventional approach to diagonalization of retrieval-based error covariance matrixes. That form of diagonalization does not reduce the rank of the rotated error covariance matrix. FYI: the diagonalization rotation is not a compression/rank reduction step. 17. page 17, line 12: 'The CPSR-RJ3 experiment skill improvement ...' How about the significance of this? The goal of Fig. 7 is to show that conventional method for assimilating a truncated retrieval profile (L10VMRR-RJ3 EX) and the phase space method (CPSR-RJ3 EX) give similar results when compared to assimilating the full retrievals (L10VMRR and CPSR respectively). The significance of the difference between L10VMRR and L10VMRR-RJ3 and that of the difference between CPSR and CPSR-RJ3 was not determined. However, the figures have been revised to include error bars generated using the ensemble samples. The L10VMRR difference may not be significant, but the difference between CPSR and

CPSR-RJ3 are likely significant. But the significance of those differences is not a key point here because: (i) we are interested in whether the extension of the CPSR method to truncated retrievals gives the expected results, (ii) it gave expected results in the upper troposphere but not in the middle and lower troposphere, and (iii) the unexpected results mean that we need to revise the extension and/or develop other methods for assimilating truncated retrievals. 18. page 17, line 21: 'Reject Top Three'. Besides the point that this explanation of the meaning of the "RJ3" experiment comes rather late, it is also not clear what justifies this method of MOPITT bias correction – given earlier publications about the nature of the bias. An explanation of RJ3 has been added to the text in Section 5.2. This is not a bias correction algorithm. It is a phase space method for not assimilating retrieval observation that are thought to be bad observations. See P18 L12 to P 18 L15. 19. Figure 4 - 5: The discussion in the text is hard to follow, because it requires comparing figure 4 and 5. Why not show model – data differences, before / after assimilation in one Figure? The discussion in the text has been revised to ease the comparison of the figures. See P16 L16 to P17 L9. 20. Figure 10: Why do these plots show results for all levels, whereas they represent experiments in which retrieval results for certain levels are not taken into account. How can these levels nevertheless show up? As explained in Section 4.3, the averaging kernel starts out as a square matrix whose dimensions depend on the dimension of the retrieval profile. In the paper, the dimension of the retrieval profile is n so the dimension of the averaging kernel is n x n. If we do not assimilate q elements of the retrieval profile, then the adjusted averaging kernel has dimensions (n – q) x n. The adjusted averaging kernel maps the true atmospheric state (observed on the n levels of averaging kernel profile) to the n – q levels of the truncated retrieval profile. Figure 10 displays the n levels of the averaging kernel profile. All n levels are present because even though the retrievals at the q levels are discarded, the retrievals at the n – q levels that are assimilated are dependent on the true atmospheric state at all n levels of the averaging kernel profile. Technical Details: 1. Title: misses a closing parenthesis (maybe better to remove the details enclosed in parenthesis anyway) Corrected. 2. page 9, line 3: remove 'changes

in' Corrected. 3. page 9, line 4: 'retrieval' i.o. 'observation' Corrected. 4. page 12, line 15: 'artifact' i.o. 'artifice' Corrected. 5. page 12, line 25 and onwards: 'sensitivity' io 'variability' Corrected. 6. page 12. line 7: 'Fig. 3' i.o. 'Fig. 2' Corrected. 7. page 16, line 4: 'Section V.C.' Corrected. 8. Figure 4, title of the lower left panel: CPSR – MET Corrected. 9. Figure 8, 9, titles of panels in the bottom row: What is 'Del-Fcst'? Corrected. RESPONSE TO REVIEWER 2 General Comments: 1. Expansion of the Introduction: The Introduction has been revised. See P1 L11 to P3 L24. 2. Additional Information for the IASI CO Retrievals: Revised. See P6 L20 to P7 L3. 3. Comparison with Observations Lacks Rigor: Figures 1, 6, and 7 have been revised to address this concern. Figures 4, 5, 8, and 9 are intended to be qualitative to show the types of changes chemical data assimilation can make. The discussion of these figures has been revised to make the comparison more clear. See P6 L20 to P7 L3. 4. Displaying Singular Vectors without Singular Values: As this comment relates to Fig. 3, we have added Table 2 which contains the singular values. As it relates to Fig. 10, Table 3 (the former Table 2) contains the singular values. The text has been revised accordingly. See P14 L21 to P14 L24; P20 L6 to P20 L23. 5. Comparison with Other Papers Assimilating MOPITT CO: The paper references some other papers that assimilated MOPITT CO. Those were related to global forecast models. There are no papers (that we know of) that have assimilated MOPITT CO in a regional model. Generally, it is not appropriate to compare the results of chemical data assimilation in a global model with that in a regional model due to the presence of lateral boundary conditions in the regional model. If for the sake of discussion, we made such a comparison, we would find that the magnitude of the improvements of the CPSR experiment compared to the MET experiment in the domain-averaged vertical profiles are comparable. See e.g., Barre et al. (2015). However, that statement is not true for the VMRR and L10VMRR experiments. The lack of an impact in those experiments is a result of our short study period and not tuning the observation errors. But as explained in the paper at P12 L 4 to P12 L9 we do not view that as a deficiency in the experimental design. We are interested in the assimilation of CPSRs. If they show an impact during a shorter study

period but more conventional methods that do not account for redundant information or error correlations fail to show an impact, then that failure identifies deficiencies in the conventional methods. The paper's point is to compare the CPSR assimilation results with independent observations, extend the CPSR algorithm to truncated retrieval profiles, and explain why assimilating truncated profiles may give unexpected results. The paper has been revised accordingly. See P11 L3 to P11 L 7. 6. The CPSR "Take Away" Message: In addition to the computational and storage efficiencies associated with the CPSR method, the "take away" message from Mizzi et al. (2016) and this paper is that the CPSR approach is the more accurate way to assimilate retrieval profiles and that failing to account for the observation error covariance and averaging kernel linear dependencies can lead to unexpected results (as illustrated in the paper by the VMRR and L10VMRR experiments which suggest that it is necessary to: (i) tune the observation error variance and (ii) use a longer study period to get an assimilation impact. The CPSR and QOR results in this paper highlight those deficiencies with the conventional method. The paper has been revised to make these "take away" messages more clear. See P11 L3 to P11 L 7; P16 L12 to P16 L14. 7. Figure 1 Suggests that Assimilating Biased Retrievals Performs Better than Not Assimilating Biased Retrievals. Is that a Reflection on How Fig. 1 Was Prepared? Are There Other Papers that Have Assimilated Truncated Retrieval Profiles? We agree that is one interpretation of Fig. 1, and it is a limitation of using a bulk verification statistic where the skill reduction in the upper troposphere is offset/dominated by the skill improvement is most of the middle and lower troposphere. 8. Another "Take Away" Message Is that the Assimilation of Raw Retrievals Shows Little Impact. The Mention of Other Papers Showing an Impact is Necessary: This is not an intended message in this paper. The paper has been revised to clarify this point. See P11 L3 to P11 L 7. Specific Comments: 1. P1L18: The choice of 'results confirm' suggests that a computational assessment is performed and included in this paper, which is not the case. It may be a matter of rephrasing and or expanding, in the introduction, on the computational benefit indicated in Mizzi et al. (2016) in use of CPSR. Not sure of your concern here. Mizzi et al. (2016) show that

when assimilating CPSRs there is a computational cost reduction due to the reduced number of observations to be assimilated. In this paper, the assimilation of CPSRs as applied to full retrieval profile necessarily has the same computational cost reductions as found in Mizzi et al. (2016). 2. P1L23-24: Point (ii) is not specifically shown in this paper. Agreed, the associated text is removed throughout. 3. P2L10: This line is a summary line of a result of Section 5.1. Might best be removed by referring to issues and concerns to be addressed in the paper and not the results themselves. Agreed, the referenced text is revised. 4. P2L12: "In the second part of the paper" refers to what section? As well it assumes a first part which has not been specified explicitly (this referring the P2L10 above). It is suggested to begin this sentence (if kept) instead with 'Therefore, we . . ..' Revised. 5. P2L13: "The rest of this paper" would best be replaced by "This paper" considering P2L12 above and that the results section is also alluded to below. Revised. 6. P2L14-18: Sections 2, 3, 4, and 5 instead of II, III, IV and V. This applies to one or two more places in the paper. Revised. 7. P1L16: '. . . and an extension of CPSRs' (added 'an') Revised. 8. P1L17: Might be worthwhile to refer here to the content of the two subsections in Section 5 and Section 2 We are unsure what is meant by subsections in Section 5 and Section 2. 9. P6L16 and P6L18: The two lines referring to Gaussian/non-Gaussian distributed errors seem to contradict each other somewhat. The Gaussian distribution refers to the L10VMRRs and the non-Gaussian distribution refers to the VMRRs (which have a lognormal distribution). 10. P6L20-21: Not clear on the value/meaning of this last sentence. Revised. 11. P6-7: Equation numbering not aligned (as would be from use of 'rightjustified') Revised. 12. P6L18-19: Point (ii) could refer to Eq. (2) and QOR to make even clearer the relationship between QOR and CPSR. Revised. 13. P6L20-P7L1 and P7L4-P7L8 are somewhat repetitive. Maybe part P6L20- P7L1 could be removed with some changes for an introduction to what follows. P6L20-P7L1 says the QORs were discussed in Mizzi et al. (2015) and explain why they are being included in this paper. P7L4-P7L8 explains where QORs come from and presents their definition. The text has been revised to facilitate revisions to the Introduction. 14. P8L20-24: As pointed out earlier, one

could point to Eq. (2) and QOR for this part. Section 5: Revised. 15. P9L13: How about the MET and CO assimilation not being coupled (or being localized?) as per P5L18. Revised. 16. P9L14 and top of Figure 1: What are the units? Maybe unitless because both are referring to log(vmr)? Do these sum up the contributions from all vertical levels? Out of curiosity, how large are these values relative to the observation and background error standard deviations? This might be useful to compare with the RMSE. The comparisons are done in retrieval space and the units are ppb. The figures have been revised to include units. Yes, the metrics are computed by summing the contributions from all vertical levels. When tuning the assimilation system, we balanced the RMSE and the total spread, so they are comparable. 17. P10L1: Due only to discarding the observation error cross-covariances and not also due (at least partly) in removing the a priori effect? Just wondering? A comparison to other papers also assimilating MOPITT CO might be pertinent here. That is correct. The a priori effect is removed for both experiments. We are not sure what comparisons you have in mind. The VMRR and L10VMRR experiments are similar to what has been done in other papers, but our study period is shorter which partly explains why they get an assimilation impact and we do not. But as explained earlier, the CPSR and QOR results in this paper show that need for a longer study period and tuned observation errors highlights the deficiencies with conventional methods. In other applications, we have run experiments similar to the VMRR and L10VRMRR experiments and found significant improvements from assimilation of MOPITT retrievals. Those results are not shown or discussed because they are not relevant to the goals of this paper. 18. P10L4 and Figure 1: Would be better to split Fig. 1 in Fig. 1 (for upper panels) and Fig. 2 (for lower panels) Revised. 19. P10L4: Use of arrows might not be best. Revised. 20. P10L4-P10L23: Would some or much of this have been stated in Mizzi et al. (2016)? If so, might be best to reduce the text. This was not discussed in Mizzi et al. (2016) because there we had found that QOR and CPSR gave similar results. See response to Reviewer 1 at Specific Comment 10. 21. P10L25-P11L8: There is mention of the increased bias with MOZAIC from CPSR and QOR, this supported also by IASI CO in

Fig. 6 and related to the MOPITT bias (also displayed by compared MOPITT and IASI in Fig. 6 – if IASI has comparatively no or less bias?) IASI CO retrievals are not known to have a systematic bias similar to that discussed for MOPITT CO retrievals in the text. The MOZAIC in situ profile observations are collected at or near urban airports. As such they are thought to be representative of a polluted urban environment. That issue is discussed in the text at P13 L12 to P13 L13 and the reason why we did not plot the lower levels of the MOZAIC profiles in Fig. 2. 22. P11L13-15: Any CO assimilation papers showing or not some impact near/at the surface? Revised. 23. P11L12: 'little or no change' instead of 'little or no improvement' as whether or not there is any improvement is not shown here. Revised. 24. P11L19: The Fig. 2 blow-up histograms are not really needed. It's up to the authors. Might it be best to split the histogram and the lower panels into two separate figures? We felt that the blow up of the histograms helped to reveal the details of the distribution that were relevant to the discussion in lines P14 L1 to P14 L14. We have not separated the histogram and singular profile figures because we want to associate the singular vectors and the DOFS histograms and we want to disassociated the singular vectors and the transformed averaging kernel profiles. 25. P12L1-4 (and beyond): Could differences in the vertical of the CO background (forecast) error variances/covariances also be a contributing factor to some degree, this depending on the assimilation setup? Having some sense of the variation in the vertical of error variances (and error correlations) might be beneficial. Would differences in background error covariances in different papers contribute to explaining differences in results? These figures/results do not depend on the background error covariance. They are an analysis of the terrestrial MOPITT CO profiles (the observations) assimilated during the study period. We agree that the vertical distribution of the observation error variance impacts these results. That aspect is addressed/discussed in the Fig. 3. Differences in the reported background error covariance do not explain these differences because they are independent of the background fields. 26. P12L12: Was any scaling really needed? P12L7-22 (and beyond): See General Comments on the display of the singular vectors. P12-P13: I only skimmed the text for the review on these pages. P13L17-18: One might question the application of the scaling in the first place. Due to the symmetry of the SVD, the singular vectors produced by the SVD subroutine and -1.0 times those singular vectors are both valid solutions to the SVD. The vertical structure of the singular vectors for each mode depends in part on the vertical distribution of the MOPITT instrument sensitivity. From the literature, we know that MOPITT has sensitivity to CO in the upper and lower troposphere. For ease of interpretation we chose a +/- 1.0 scaling (for the first and second rows as discussed in the paper) that make the singular vector vertical sensitivity consistent with that published in the literature. Then we applied that scaling consistently throughout the discussion in this section. 27. P13L18: e.g. '. . . that, when . . . is considered, the . . .' (while this is likely somewhat subjective, adding some commas here and or similarly elsewhere in the paper might be considered) Revised. 28. P13L20: '. . . and the first . . .' (added 'the') Revised. 29. P13L22: Might the validity of this assertion depend on the singular values? For the diagonalization transform, the modes are ordered by decreasing singular value. The singular value is the compressed observation error variance after accounting for the covariance. When we project the compressed averaging kernel onto the singular vectors of the compressed retrieval error covariance matrix as scaled by the square root of corresponding singular value, the result shows the vertical sensitivity of the transformed averaging kernel removing linear dependences and after accounting for the: (i) error covariance terms, (ii) the magnitude of the observation error variance, and (iii) the vertical structure of the observation error covariance. When the modes are ordered by decreasing singular value, the referenced statement is valid. 30. P14L2-3: Please indicate actual references and elaborate on results where applicable. Revised to add references. 31. P14L3: What is meant by 'do not adjust for the averaging kernel linear dependencies or for the observation error covariance"s'". [Might the latter be in reference to not including error correlations (cross-covariances)?] The CPSR algorithm perform two tasks: compression and diagonalization. The compression task compresses the averaging kernel by removing linear dependencies. The diagonalization task diagonalizes the compressed

covariance matrix by rotating the compressed quasi-optimal retrieval equation into a coordinate space that maximizes the compressed variance (thereby accounting for the covariance). Since no other chemical data assimilation researcher is compressing the averaging kernel and rotating the compressed observation error covariance matrix, they are not adjusting for the averaging kernel linear dependencies or for the observation error covariance. This is what is meant. 32. P14L11-13 (and remainder of the paragraph): While there is some level of consistency in the coastal regions, it is not that evident that one could say that the analysis and forecasts are 'generally consistent' with the observation. Maybe some re-phrasing would be needed. A quantitative evaluation might help. The text has been revised to make the comparisons and conclusion more clear. 33. P14L15-16: Has (i) been looked at to some degree? The text has been revised and this has been removed. 34. P14L16:17: Has (ii) been verified? No, because there is no way to determine whether the reported emission are too low. 35. P14L17: The changes in the analyses seem rather weak in the central U.S. or there- abouts in comparison what is needed to increase the analysis to levels fairly close to what is seen in Fig. 5. Might a quantitative evaluation help? The areal coverage of IASI is greater than that of MOPITT. In the central US where there are MOPITT observations, there is increased CO so that the magnitudes are in better agreement with IASI. Where there are no MOPITT observations, there is decreased CO so that the magnitudes are in worse agreement with IASI. 36. P14L20: Does this refer to the central U.S. or is an overall assertion? It is not so clear from the figures if for the central U.S.. Either way, a quantitative evaluation (by regions maybe) might be more meaningful to justify this assertion (and those above). This discussion has been revised to make it slightly more quantitative. We are reluctant to make it too quantitative because the point of Figs. 4, 5, 8, and 9 is to show examples of: (i) the types of horizontal impacts one gets from chemical data assimilation, and (ii) how those impacts correspond to the observations. The text has been revised to make this more clear. 37. P14L25: Might it be worth to mention/discuss the level of similarity and differences between 'SS' and 'RS' profiles? The figures have been revised to remove the state space profiles. 38. P15L3-6: 'for

pressures less than about 500 hPa, the MOPITT CO assimilation with CPSR draws the forecast and analysis further away from IASI while the opposite occurs for larger pressures.' Could refer to the comparison to MOZAIC in Fig. 1 to support the comparison with IASI in the upper levels. For pressures less than 250 hPa, the CPSR experiment draws the forecast and analysis further away from IASI in Fig. 6. The text has been revised accordingly. 39. P15L11-21: An alternative would be for a version of this 'summary' to instead be in the 'Summary and Conclusions' section. It's up to the authors. We decided to leave this here as an intermediate conclusion. 40. P15L14: It is not really that the 'phase space' observations error variances is reduced as oppose to the transformation allowing to account for the otherwise neglected 're- trieval space' error correlations. The CPSR experiment has better verification scores compared to VMRR and L10VMRR because the: (i) correlations are greater, and/or (ii) transformed observation errors are smaller. We think it is primarily due to smaller observation errors. They are smaller due to the compression step in the CPSR transform. They cannot be smaller due to the diagonalization step because that is a variance maximizing rotation. If the compression step had no filtering effect on the errors, then the variance resulting from the diagonalization step would no smaller than that from the compression step. 41. P15L16-17: Part of (ii) is actually a repetition of (i). Some change in the sentence is needed. Revised. 42. P15L17: As part of (ii), has the statement 'linearly dependent portion of the transformed retrievals do not . . .' (repeated earlier as well) been verified, noting that background error covariances (and its non-zero error correlation coefficients) contribute to determining the distribution of information for strongly overlapping averaging kernels (likely requiring more computational effort though). Any other references for his part (e.g. Migliorini, 2008 and or 2012 or even Mizzi 2016)? If so, they should also be indicated earlier on in this paper. The statement has been verified because the QOR algorithm is now part of the CPSR algorithm. So, the same computer code is used for the QOR results as is used for the QOR part of the CPSR results. Thus, the similarity of results for the CPSR and QOR experiments implies that the linearly-dependent part of QOR profile remaining after the transformation does not

contribute to the analysis increment. No other researchers (that we know about) have found this result. 43. P15L21-23: Have other assimilation studies shown this as well – that the resulting CO analyses and forecasts in the upper levels would be biased. This result would be expected considering the literature on the MOPITT CO data – assuming IASI and also MOZAIC CO is less biased. Might be good to indicate that this was not entirely un expected. The only prior study that made this observation (of which we are aware) was cited previously Barre et al. (2016). Note: Barre et al. (2016) did not present results from assimilating the biased retrievals. They discarded them before performing their forecast/assimilation experiments. 44. P15L23. This also applies to the comparison with MOZAIC CO. Same response as in 42. 45. P16L4: Section V.C to be changed. Corrected. 46. P16L8: '. . .accounts for the error correlations of the observation error covariance matrix.' We agree that the error covariance and error correlations are related. In the paper and in the CPSR algorithm we are accounting for the error covariance. We have used that terminology consistently throughout the paper. So, we have not made this change. 47. P16L11: e.g. 'that, in the upper troposphere, the' (commas) P16L17: e.g. 'troposphere, there' Revised. 48. P16L18: 'A comparison with' Revised. 49. P16L21-22: Could be re-phrased. Revised. 50. P17L1: Remove 'However', i.e., 'The forecast . . .' Revised. 51. P17L3: '... United States similar to, though weaker than, the CPSR experiment' or something similar Revised. 52. P17L5-7: e.g., 'The upper tropospheric impacts of Fig. 9 show even smaller changes for the CPSR-RJ3 experiment except for the reductions over the southeastern United States. The CPSR-RJ3 experiment therefore further demonstrates, in addition to Fig. 7, the reduction of bias in the upper troposphere through the removal of the biased observation profile elements, this though at the expense of reduced improvements in the lower troposphere.' Revised. 53. P17L9-11; This should explicitly refer to the upper right-hand side panel with the comparison to IASI CO. This refers to both the MOPITT and IASI comparisons. The text has been revised. 54. P17L11: The improvement is rather small though as compared to CPSR (and QOR) in Fig. 1. This needs to be indicated. Is this related to how this diagnostic is generated, e.g. maybe because of

a dominance of the lower tropospheric RMSE contributions (as compared to the upper layers)? Yes, this is related to how the metric in Fig. 1 is generated. The text has been revised to indicate that the improvement is slight. 55. P18L5-6: This should refer to the levels with pressures below about 500 hPa. 'Significantly' seems to be an exaggeration based on the curve. I suggest removing 'significantly'. \ Revised. 56. P18L7-13 and Figure 10 (with Table 2): The bottom row of Fig. 10 (even in combination to Table 2) suggests that the 'reject middle three' may have least impact in assimilation. This would be contrary to just looking at the traces in Table 2 which, based on the earlier statement, indicate the 'reject bottom three' provide the least amount of info. Am I missing something? Any discussion or comments. Table 3 (the former Table 2) indicates the amount of variability explained (or the amount of information remaining) after discarding (not assimilating) the referenced levels. For the trace, this table indicates that: (i) the Full Profile contains the most amount of information, (ii) Mode 1 explains .9639/1.452=66% of the variability, (iii) Mode 2 explains 33%, and (iv) Mode 3 explains 1%. With that interpretation, rejecting the top three levels has the greatest reduction in the total information and rejecting the bottom three levels has the least reduction. The bottom row of Fig. 11 (the former Fig. 10) shows the vertical sensitivities for each mode. For the Full Profile (column (a)) most of the maximum sensitivity is near 250 hPa and the magnitude is ∼6. For rejection of the top three levels (column (b)), the maximum sensitivity is between 350 hPa and 700 hPa, and the magnitude is ∼2. It is those changes in the total information and vertical sensitivity that explain the results in Fig. 7 (the former Fig. 6). 57. P19L9: Removing 'likely' seems appropriate as it seems pretty certain. Revised. 58. P19L16: Instead of 'magnitude of the observation errors' is it more the omission of the 'observation error correlations in the assimilation' in comparison the CPSR and QOR effects? We do not agree with this interpretation. See responses to Reviewer I, General Comment 9 and Reviewer II, Specific Comment 40. 59. P19L19: 'Truncated the observation errors' may not be the correct wording considering the above. Please see response to Reviewer II, Specific Comment 57. 60. P19L22-23: Applies also to IASI CO. Even better would be to instead mention

MOZAIC CO at P20L5. Revised. 61. P20L2: 'because, by accounting for . . . error correlations ,' We account for observation error covariance. While it is true that these indicate observation error correlations, we are explicitly accounting for the covariance (not the correlations). 62. P20L21-P21L3: Different contradictory statements in this sentence related to the impact at the surface. Also, one could mention the approximate proportion of cases where surface impact may occur. One might also consider the background error variances in also contributing to the level of impact at near the surface (on top of the averaging kernels themselves (and obs error covariances)) We were unable to identify the contradictory statements in the lines listed which are in the Code and Data Availability section. 63. P20L10: 'confirming the applicability of the CPSR . . .' Revised. 64. P20L13: 'Excluding the assimilation of some elements of the observation profiles can ...' Revised. 65. P20L16: 'to address the reduced impact from not assimilating retrieval profile levels' ('reduced' instead of 'remote' and . . .) Revised. 66. Table 1: Might be better to follow the form of Table 1 in Mizzi et al. (2016) Revised. 67. Figures: Font sizes for panels with y-axis as pressure are on the edge of being too small or are too small. Please check. Revised. 68. Figures 6 and 7: For clarity, might be best to drop the 'SS' results (at least for Fig. 7 if not both). That is unless the one intends to mention and discuss in the text, for Fig. 6 for example, the level of similarity and differences between 'SS' and 'RS'. Revised. 69. Figure 10: 'except that this figure' (added 'that') Revised. 70. Figure 7 (lower panels): Unless this is a visual clarity issue, it seems that the Met EX RS results near the surface differ between the CPSR panels and the L10VMMR panels, while they would be expected to be the same. Please check. The Met EX RS results are not the same for the L10VMRR and the CPSR experiments. For the L10VMRR experiment the retrieval equation (the retrieval equation is used to map the state space CO profile into retrieval space) is the full equation. For the CPSRRJ3 experiment, the mapping is based on the truncated retrieval equation (the equation after discarding the biased elements, rows, and columns as appropriate).

---

## Author Comment (AC2) · 10 Aug 2018

[revised manuscript text omitted]

$U_0^T(y_r - (I - A)y_a - \varepsilon) = S_0 V_0^T y_t$                                                         (4)

and the compressed error covariance is

$U_0^T E_m U_0$.                                                                (5)

In that step, there is no filtering of the dominate modes. Next, we apply the diagonalization transform. If the SVD of the compressed error covariance in (5) is $U_0^T E_m U_0 = \Phi \Sigma \Psi^T$, then the diagonalized and conditioned form of Eq. 4 is

$\Sigma^{-1/2} \Phi^T U_0^T (y_r - (I - A)y_a - \varepsilon) = \Sigma^{-1/2} \Phi^T S_0 V_0^T y_t$                           (6)

[revised manuscript text omitted]